

# Organic matter across subsea permafrost thaw horizons on the East Siberian Arctic Shelf

Birgit Wild[1,2], Natalia Shakhova[3], Oleg Dudarev[3,4], Alexey Ruban[3], Denis Kosmach[4], Vladimir Tumskoy[5,6,7], Tommaso Tesi[8], Hanna Joß[1], Helena Alexanderson[9], Martin Jakobsson[2,10], Alexey Mazurov[3], Igor Semiletov[3,4], Örjan Gustafsson[1,2]

[1]Department of Environmental Science and Analytical Chemistry, Stockholm University, Stockholm, 11418, Sweden
[2]Bolin Centre for Climate Research, Stockholm University, Stockholm, 11418, Sweden
[3]Tomsk Polytechnic University, Tomsk, 634050, Russia
[4]Pacific Oceanological Institute, Russian Academy of Sciences, Vladivostok, 690041, Russia
[5]Moscow State University, Moscow, 119991, Russia
[6]Institute of Geography, Russian Academy of Sciences, Moscow, 119017, Russia
[7]University of Tyumen, Tyumen, 625003, Russia
[8]Institute of Marine Sciences, National Research Council (ISMAR-CNR), Bologna, 40129, Italy
[9]Department of Geology, Lund University, Lund, 22361, Sweden
[10]Department of Geological Sciences, Stockholm University, Stockholm, 11418, Sweden

*Correspondence to*: Birgit Wild (birgit.wild@aces.su.se), Örjan Gustafsson (orjan.gustafsson@aces.su.se)

**Abstract.** Thaw of subsea permafrost across the Arctic Ocean shelves might promote the degradation of organic matter to $CO_2$ and $CH_4$, but also create conduits for transfer of deeper $CH_4$ pools to the atmosphere and thereby amplify global warming. In this study, we describe sedimentary characteristics of three subsea permafrost cores of 21-56 m length drilled near the current delta of the Lena River in the Buor-Khaya Bay on the East Siberian Arctic Shelf, including content, origin and degradation state of organic matter around the current thaw front. Grain size distribution and optically stimulated luminescence dating suggest the alternating deposition of aeolian silt and fluvial sand over the past 160 000 years. Organic matter in 3 m sections across the current permafrost table was characterized by low organic carbon contents (average $0.7 \pm 0.2\%$) as well as enriched $\delta^{13}C$ values and low concentrations of the terrestrial plant biomarker lignin compared to other recent and Pleistocene deposits in the study region. The lignin phenol composition further suggests contribution of both tundra and boreal forest vegetation, at least the latter likely deposited by rivers. Our findings indicate high variability in organic matter composition of subsea permafrost even within a small study area, reflecting its development in a heterogeneous and dynamic landscape. Even with this relatively low organic carbon content, the high rates of observed subsea permafrost thaw in this area yield a thaw-out of 1.6 kg OC m$^{-2}$ year$^{-1}$, emphasizing the need to constrain the fate of the poorly described and thawing subsea permafrost organic carbon pool.



# 1 Introduction

More than two thirds of subsea permafrost is thought to be located on the East Siberian Arctic Shelf (ESAS), the World's largest and shallowest continental shelf sea that comprises the Laptev, East Siberian and Russian Chukchi Seas (Fig. 1). During the late Pleistocene, the sea level was lower than today, the ESAS was exposed to the atmosphere and accumulated thick permafrost deposits (Nicolsky et al., 2012; Romanovskii et al., 2004). Part of the Pleistocene permafrost was immediately destroyed by erosion when sea levels started to rapidly rise after the Last Glacial Maximum (Romanovskii et al., 2000), while the remainder was inundated as subsea permafrost. This process is still ongoing, as indicated by coastal erosion rates of up to 5 m year-1 along the ESAS (Lantuit et al., 2012). Subsea permafrost on the ESAS is likely thawing both from below via geothermal heat flow since the region is tectonically highly active (Nicolsky et al., 2012; Romanovskii and Hubberten, 2001), and from above due to the thermal gradient from overlying seawater (Nicolsky et al., 2012; Shakhova et al., 2017). The current thermal state of subsea permafrost and its future trajectory are poorly understood, but crucial for the Arctic carbon budget.

Permafrost deposits store vast amounts of organic carbon that is largely protected from microbial degradation while frozen, but might be converted to the greenhouse gases $CO_2$ and $CH_4$ when thawed and thereby eventually further strengthen global warming. While organic carbon stocks and greenhouse gas emissions of terrestrial permafrost are progressively getting better constrained (e.g., Harden et al., 2012; Hugelius et al., 2014; Schuur et al., 2015), subsea permafrost is more difficult to access and therefore remains one of the largest uncertainties concerning natural greenhouse gas emissions over the coming decades and centuries. Field campaigns to the ESAS have observed a strong supersaturation of $CH_4$ in seawater above subsea permafrost over large areas compared to the atmosphere (Shakhova et al., 2010b, 2014, 2015). A current key challenge is to constrain the relative contribution of $CH_4$ from different plausible subsea sources (e.g., Overduin et al., 2015a; Sapart et al., 2017; Shakhova et al., 2010b), including (i) microbial degradation of thawing subsea permafrost organic matter, as well as (ii) release of $CH_4$ stored within subsea permafrost, from shallow $CH_4$ hydrates that have been proposed for the ESAS (Romanovskii et al., 2005), and from thermogenic/petrogenic $CH_4$ pools at great depths underneath (Cramer and Franke, 2005). The latter pools are likely trapped within and underneath frozen subsea permafrost, but may escape to the surface as permafrost thaws and gas migration pathways form (Romanovskii et al., 2005; Shakhova et al., 2010a, 2010b). The large extent of subsea permafrost on the Arctic Ocean shelves and its potential as a source, or physical blockage, of greenhouse gas transport to the atmosphere highlight the need to constrain the quantity and quality of organic carbon currently stored in subsea permafrost, and its fate upon thaw.

This study focusses on subsea permafrost in the Buor-Khaya Bay in the south-eastern Laptev Sea on the ESAS where particularly high rates of thaw have been recently reported by redrilling subsea permafrost at several sites three decades apart (Shakhova et al., 2017). Comparison of the ice-bonded permafrost table (IBPT) in 1982/83 and 2013/14 suggests an average IBPT deepening of $14 \pm 3$ cm year$^{-1}$ in the Buor-Khaya Bay (Shakhova et al., 2017). Given the current permafrost table at 10-20 m depth, the deepening of the permafrost table by about 4 m over the last three decades suggests a very recent warm-up of subsea permafrost to the thaw point. The now rapid thawing of subsea permafrost is expected to continue and possibly intensify



in the future if water temperatures in these shallow waters continue to rise with anthropogenic warming. Based on a unique set of three 21-56 m long cores that were drilled from the fast ice in the Buor-Khaya Bay in 2012 and 2013, we here present the quantity, origin and degradation state of organic matter around the current thaw front along with a lithological description and chronology derived from optically stimulated luminescence dating (OSL; n = 6).

## 2 Material and Methods

### 2.1 Study area

The Buor-Khaya Bay is part of the Laptev Sea and the ESAS, located south-east of the Lena River delta (Fig. 1). The area has been affected by marine re- and transgressions during the Pleistocene and Holocene. The ESAS was exposed to the atmosphere already before the MIS 5e interglacial (Kienast et al., 2008, 2011), i.e. at least 130 ka before present (Andreev et al., 2011), and was then part of Beringia, a continuous, non-glaciated land mass that encompassed Eastern Siberia, Alaska, and Western Canada (Anderson and Lozhkin, 2001; Murton et al., 2015). Large amounts of organic carbon accumulated in Beringia during the Pleistocene, often in the form of ice-rich, fine-grained permafrost deposits known as Ice Complex deposits or Yedoma (Romanovskii et al., 2000; Schirrmeister et al., 2011c; Zimov et al., 2006), but also as fluvial and alluvial deposits. These deposits were partly degraded by thermokarst formation during warm periods in the MIS 5e interglacial (Schirrmeister et al., 2011b) and at the end of the Pleistocene (Rekant et al., 2015; Romanovskii et al., 2000), and later by thermal, sea-ice and wave-induced erosion when the ESAS became submerged after the Last Glacial Maximum (Bauch et al., 2001; Overduin et al., 2016; Rekant et al., 2015). While Pleistocene permafrost deposits are at least partly preserved on land (Schirrmeister et al., 2011b; Strauss et al., 2013), it is unclear how much of the original permafrost on the ESAS still exists (Rekant et al., 2015; Romanovskii et al., 2000).

### 2.2 Field work

Subsea permafrost drill cores for this study were obtained during expeditions in spring 2012 and 2013. Cores were drilled from the fast ice in the Buor-Khaya Bay, using a drilling rig (URB-4T) with a hydraulic rotary-pressure mechanism that operates without drilling fluid to avoid carbon contamination. Well tubes and borehole casings were 4 m long and 147 mm in diameter. Casings were drilled into the seabed, and cores were extracted from the bore holes, sectioned and transported to Tiksi in thermo-insulated boxes for storage at -12°C. Details on drilling procedure and logistics have been described by Shakhova et al. (2017).

The three cores analyzed in this study (4D-13, 2D-13, 4D-12) were drilled on the subsea thermo-erosion terrace of Muostakh Island (71-72°N, 129-130°E; Fig. 1). Water depths at the drill sites were between 2.5 and 3.4 m in 2012/13, following submersion 145-511 years before present at this near-coastal location (Shakhova et al., 2017; Table 1). All three cores crossed through the IBPT, i.e., the upper part was thawed and the lower part frozen at the time of sampling. Core 4D-13 was 21.0 m long, the IBPT was located at 8.9 m depth, and IBPT deepening rates of $18.3 \pm 0.1$ cm year$^{-1}$ have been observed between





1982 and 2013 (Shakhova et al., 2017). Core 2D-13 was 30.4 m long, with the IBPT at 15.9 m, and IBPT deepening rates of $9.3 \pm 1.7$ cm year$^{-1}$ (Shakhova et al., 2017). Core 4D-12 was 55.7 m long, with the IBPT at 23.9 m. Rates of IBPT deepening have not been measured at this site. Grain size distribution was determined with a laser microanalyser (Analysette 22, Fritsch) for fine-grained and sieve analysis for coarse-grained sediments (Dudarev et al., 2006). A detailed method description as well

as original data have been published by Shakhova et al. (2017). Statistical properties of grain size distributions were calculated using Gradistat v8 (Blott and Pye, 2001) and results are provided in Table S2. In addition to a lithological description of the three cores, we here present an OSL-derived chronology for core 4D-12, and a detailed analysis of organic matter at the current thaw front of subsea permafrost. To that end, high-depth resolution subsamples of the 3 m sections extending across the IBPT of each core (4D-13: 7-10 m depth; 2D-13: 14-17 m depth; 4D-12: 22-25 m depth) were thawed at room temperature and

manually homogenized, and aliquots were freeze-dried before analysis of specific surface area, bulk organic matter properties and lignin phenol concentrations. Lignin is a biopolymer produced by terrestrial higher plants and ratios between individual lignin phenols serve here as proxies of origin and decomposition state of subsea permafrost organic matter.

**2.3 Optically stimulated luminescence chronology**

Samples for OSL dating were processed under dark room conditions. Subsamples of core 4D-12 were taken by pushing a tube

into the drill core that contained frozen sediments. The tubes were opened at the Lund Luminescence Laboratory, Sweden, samples were wet sieved to extract the 180-250 µm fraction (90-180 µm for one sample at 15 m depth due to its finer-grained nature) and subsequently treated with 10% HCl, 10% $H_2O_2$, 38% HF (60 min.) and a second time with 10% HCl. Density separation with LST Fastfloat at 2.62 g cm$^{-3}$ was used to separate quartz from feldspar grains before HF etching. After the final HCl treatment, extracts were re-sieved at 180 µm (90 µm for one sample at 15 m depth).

Dose measurements were carried out on large aliquots in a Risø TL/OSL reader model DA-20 with a $^{90}Sr/^{90}Y$ beta radiation source (~0.15 Gy s$^{-1}$). Single Aliquot Regeneration protocol (Ankjærgaard et al., 2010; Banerjee et al., 2001; Murray and Wintle, 2000, 2003) settings were determined individually for each sample based on infrared/blue ratios, dose recovery and preheat plateau tests (Table S1). Most samples suffered from some apparent feldspar contamination (high infrared/blue ratios) and had relatively dim signals and significant scatter in equivalent doses. A fast signal component was nevertheless present

and dominating, and dose recovery tests showed that the analytical protocols gave accurate results (mean ratio $0.97 \pm 0.05$, n = 15). However, two samples, from 35 and 47 m depth, were rejected due to very poor dose recovery ratios and not measured further (Table S1). Aliquots were accepted if they had a recycling ratio <10% from unity and a test dose error <10% (<20% for samples at 17, 42, and 51 m depth). Equivalent doses were calculated in Risø Analyst, using exponential curve fitting, and the Central Age Model (Galbraith et al., 1999) was applied.

The sediment dose rate was determined by gamma spectrometry at the Nordic Laboratory for Luminescence Dating in Denmark (Murray et al., 1987). Field and saturated water content were measured by weighing subsamples in the laboratory, and the average water content since time of deposition was estimated as having been saturated for 90% of the time and





unsaturated (water content as when opened in lab) during 10% of the time. Environmental dose rates and final ages were calculated in the DRAC online calculator (Durcan et al., 2015).

## 2.4 Specific surface area of minerals

The specific surface area of minerals was determined as described by Bröder et al. (2016b). Briefly, freeze-dried samples (n = 56) were combusted at 400°C for 12 h to remove organic matter, rinsed twice with Milli Q water, freeze-dried again and degassed under $N_2$ flow at 200°C for 2 h using a Micromeritics FlowPrep 060 Sample Degas System. Surface area was analyzed with a Micromeritics Gemini VII Surface Area and Porosity Analyzer using the Brunauer-Emmett-Teller technique (Brunauer et al., 1938), $N_2$ as absorbent and six adsorption isotherm points for calibration. System performance was monitored with black carbon and titanium reference material provided by Micromeritics.

## 2.5 Bulk organic matter properties

For analysis of organic carbon and total nitrogen concentrations as well as $\delta^{13}C$ values of organic carbon, freeze-dried subsamples were ground in a mortar, and aliquots were filled into Ag capsules, acidified with 1 M HCl and dried to remove carbonates. The acidification procedure was repeated until effervescence stopped. Samples were analyzed using a Finnigan Delta Plus XP mass spectrometer coupled to a Thermo Fisher Scientific Flash 2000 IRMS Element Analyzer via a Conflo II interface. Analytical uncertainty was determined for a subset of samples in triplicates (21 of 153 samples). Standard deviations for the triplicates averaged 0.039% of sample dry weight for organic carbon, 0.003% of sample dry weight for total nitrogen, and 0.158‰ for $\delta^{13}C$. Values of individual samples are presented in Table S3.

## 2.6 Biomarker analyses

Micro-wave assisted CuO oxidation of freeze-dried and ground samples was used to hydrolyze the macromolecules that constitute the bulk of organic matter, and analyze the derived lignin phenols, hydroxybenzenes and p-hydroxybenzenes (Goñi and Montgomery, 2000). Briefly, sample aliquots were amended with CuO, ferrous ammonium sulfate, and NaOH and oxidized in $O_2$-free atmosphere in an UltraWAVE Milestone microwave. After oxidation, we added known amounts of ethyl vanillin and trans-cinnamic acid as internal standards, acidified samples with HCl to pH 1 and extracted twice with ethyl acetate. Excess water was removed with anhydrous $Na_2SO_4$, and the solvent was evaporated in a centrifugal evaporator under reduced pressure at 60°C. Samples were dissolved in pyridine and stored frozen until analysis, and then derivatized with bis-trimethylsilyl-trifluoroacetamide (BSTFA) + 1% trimethylchlorosilane (TMCS). The CuO oxidation products were analyzed with a gas chromatography-mass spectrometry (GC-MS) system (Agilent 7693 autosampler, 7820A GC, 5977E MSD) with splitless injection on a DB1-MS column (30 m x 250 µm; 0.25 µm film thickness), with 1.2 ml min⁻¹ helium as carrier, an initial temperature of 50°C for 5 min, a ramp of 10°C min⁻¹ to 300°C, and constant temperature for 8 min. We quantified vanillin (Vl), acetovanillone (Vn), vanillic acid (Vd), syringaldehyde (Sl), acetosyringone (Sn), syringic acid (Sd), p-coumaric acid (pCd), ferulic acid (Fd), benzoic acid (Bd), m-hydroxybenzoic acid (m-Bd), 3,5-dihydroxybenzoic acid (3,5-Bd), p-





hydroxybenzaldehyde (Pl), p-hydroxyacetophenone (Pn), and p-hydroxybenzoic acid (Pd) against external standard curves of each compound that were measured together with the samples. The sum of vanillyl phenols (V) was calculated as Vl + Vn + Vd, the sum of syringyl phenols (S) as Sl + Sn + Sd, the sum of cinnamyl phenols (C) as pCd + Fd, and the sum of all lignin phenols as V + S + C. Total p-hydroxybenzenes were calculated as the sum of Pl + Pn + Pd. Individual values are presented in Table S4.

Six ratios were calculated to describe origin and decomposition state of organic matter. Ratios of syringyl over vanillyl subunits (S/V) and cinnamyl over vanillyl subunits (C/V) reflect the source of lignin phenols, with higher S/V ratios in angiosperm than in gymnosperm tissues, and higher C/V ratios in non-woody (e.g., leaves) than in woody tissues (Hedges and Mann, 1979). Ratios of p-hydroxybenzenes over vanillyl (P/V) served as indicators of peat since p-hydroxybenzenes are particularly abundant in Sphagnum mosses (Williams et al., 1998). Ratios of acids over aldehydes of syringyl and vanillyl subunits (Sd/Sl, Vd/Vl) reflect the decomposition state of organic matter. Both Sd/Sl and Vd/Vl typically increase during aerobic decomposition due to oxidation (Ertel and Hedges, 1984; Hedges et al., 1988; Thevenot et al., 2010). Similarly, ratios of 3,5-dihydroxybenzoic acid over vanillyl (3,5-Bd/V) increase during decomposition since vanillyl is degraded faster than 3,5-dihydroxybenzoic acid (Dickens et al., 2007; Prahl et al., 1994).

## 2.7 Statistical analyses

Statistical analyses were performed using R 3.3.1 (R Development Core Team, 2016) with the additional package GenABEL (Aulchenko et al., 2007). We tested correlations between measured parameters across the three cores using Spearman's rank sum correlations. This method tests for a monotonous relationship between two parameters; the closeness of this relationship is then described by Spearman's correlation coefficient rho. Differences between core segments above and below the IBPT were tested individually for each core using Student's t-test, after transformation where necessary to meet the conditions of the test. Spearman's rank sum correlations were further applied to test for monotonous changes with distance from the IBPT in the thawed part of the core that might reflect progressing decomposition upon thaw.

## 3 Results and discussion

### 3.1 Lithological characteristics and deposition age of subsea permafrost in the Buor-Khaya Bay

The marine transgression at the onset of the Holocene flooded a complex permafrost landscape with intact and thermokarst-affected Ice Complex deposits, lakes, and rivers that had developed over the course of the Pleistocene (Romanovskii et al., 2000). Optically stimulated luminescence dating shows that core 4D-12 reflects more than 160 ka of local deposition history (Fig. 2). Although this period covers a considerable range in climatic conditions, a close correlation of age and depth between $162 \pm 22$ ka at 51 m depth and $51 \pm 5$ ka at 17 m depth ($R^2 = 0.98$) suggests rather constant deposition at least at the low temporal resolution available, followed by a jump to $8.5 \pm 0.6$ ka at 15 m depth that might reflect a period of low deposition or an erosional event. Along the entire core 4D-12, alternations between silt- and sand-dominated deposits with mostly



unimodal grain size distributions (Fig. 2, Table S2) suggest fluctuations between predominant wind- and water-related deposition, respectively, the latter likely by rivers. Core 2D-13 showed similar changes between silt and sand layers, whereas core 4D-13 – located most closely to Muostakh Island – was characterized by largely bimodal grain size distributions with strong contributions of both silt and sand fractions at the same depth. By comparison, previous studies have described almost

exclusively sandy deposits in a subsea permafrost core even closer to Muostakh Island (Shakhova et al., 2017), as well as in a core drilled close to the eastern coast of the Buor-Khaya Bay (Overduin et al., 2015a). Overall, pronounced differences in grain size distribution within and between cores indicate high temporal and spatial variability of deposition regimes even within the small study area of the Buor-Khaya Bay.

Terrestrial sites in the vicinity of the Buor-Khaya Bay show a characteristic sequence of deposits formed under different

environmental conditions in the late Pleistocene and Holocene. Holocene deposits at the surface reach not more than a few meters thickness. Underneath, Ice Complex deposits that accumulated ca. 10-50 ka before present can extend over dozens of meters, and are characterized by high ice content as well as often multimodal grain size distribution with peaks in the silt and fine sand fractions (Schirrmeister et al., 2011c; Strauss et al., 2015). Underlying the Ice Complex deposits, coarser grained fluvial/alluvial sands from the MIS 4/5a-d (50-110 ka before present) are frequently observed, as well as thermokarst deposits

formed during the MIS 5e interglacial (115-130 ka before present), and older Ice Complex deposits from the MIS 6/7 stadial (130-200 ka before present; Schirrmeister et al., 2011a, 2011b; Slagoda, 2004). Since Ice Complex deposits along the coasts of eastern Siberia do not extend to more than a few meters below the current sea level (Schirrmeister et al., 2011b, 2011c; Slagoda, 2004), previous studies have suggested the wide-spread destruction of Ice Complex deposits on today's ESAS by thermokarst formation at the end of the Pleistocene, followed by erosion during the Holocene marine transgression (Martens

et al., in review; Rekant et al., 2015; Romanovskii et al., 2000). The alternations of unimodally distributed silt and sand in cores 2D-13 and 4D-12 are in line with the hypothesis that subsea permafrost represents mostly older deposits preserved under the eroded Ice Complex deposits. The bimodal grain size distribution of core 4D-13 – shallowest and closest to Muostakh Island – is more similar to those observed in Ice Complex deposits (Schirrmeister et al., 2011c; Strauss et al., 2015). Nevertheless, the comparatively young OSL ages measured at great depths of core 4D-12 contrast previously determined

radiocarbon ages of > 40 ka in Ice Complex deposits above sea level around the Buor-Khaya Bay (Grosse et al., 2007; Schirrmeister et al., 2002, 2011c; Strauss et al., 2015) and highlight the limitations of our current understanding of Pleistocene permafrost deposits in western Beringia.

Since this study focuses on organic matter at the current thaw front, we selected samples from 3 m sections extending above and below the IBPT of each core for analysis of organic matter properties at high depth resolution. Linear interpolation of

OSL-dates places the 22-25 m interval studied in core 4D-12 between 65 and 72 ka, i.e., in the MIS 4/5a-d. For this time frame, a cooler and drier climate than today and dominant tundra steppe vegetation have been suggested for the Laptev Sea region (Andreev et al., 2011). The corresponding thaw front sections of cores 4D-13 (7-10 m) and 2D-13 (14-17 m) are shallower than that of 4D-12; we therefore expect younger ages.





## 3.2 Bulk organic carbon at the thaw front of subsea permafrost in the Buor-Khaya Bay

The current thaw front of subsea permafrost in this area was characterized by organic carbon contents of on average 0.73 ± 0.23% (mean ± standard deviation; total range 0.37-2.62%; n = 153; Fig. 3). This range is similar to that of a previously described subsea permafrost core from the Buor-Khaya Bay (Overduin et al., 2015a), as well as to fluvial/alluvial sediments

deposited during the Pleistocene and preserved at terrestrial sites in eastern Siberia (Schirrmeister et al., 2011b; Fig. 4). Pleistocene Ice Complex Deposits, thermokarst and lake sediments in terrestrial eastern Siberia (Schirrmeister et al., 2011b), as well as recent marine sediments from the Buor-Khaya Bay (Vonk et al., 2012) show on average higher organic carbon contents (Fig. 4). Normalized by the specific surface area of minerals, average organic carbon contents of the subsea permafrost samples corresponded to 0.70 ± 0.39 mg OC m$^{-2}$ with a total range of 0.31-2.29 mg OC m$^{-2}$ (n = 56). Most values are thus

within the typical range of marine sediments of 0.4-1.0 mg OC m$^{-2}$ (Blair and Aller, 2012; Fig. 5a) and are consistent with measurements in other Arctic deposits, including recent marine surface sediments from the Buor-Khaya Bay (0.7-2.2 mg OC m$^{-2}$; Karlsson et al., 2015) and other areas of the Laptev Sea (0.3-2.2 mg OC m$^{-2}$; Bröder et al., 2016b) and the East Siberian Sea (0.3-1.0 mg OC m$^{-2}$; Bröder et al., 2016a; Karlsson et al., 2015), recent sediments of the Mackenzie River and adjacent lakes (0.7-1.5 mg OC m$^{-2}$ and 0.6-1.2 mg OC m$^{-2}$, respectively; Vonk et al., 2015), as well as recent mineral active layer soils

in Western Siberia (0.2-0.4 mg OC m$^{-2}$; Buchkina et al., 1998) and on Spitsbergen (0.7-2.9 mg OC m$^{-2}$; Cieśla et al., 2018). We conclude that the low organic carbon content of the subsea permafrost samples in this region was partly related to a low capacity of rather coarse-grained minerals to bind organic carbon.

Eastern Siberia is dominated by C3 vegetation with $\delta^{13}$C values between -25‰ and -30‰ (O'Leary, 1988). The $\delta^{13}$C values of recent and Pleistocene organic carbon deposits in Eastern Siberia are in the same range; average values of thermokarst and

lake deposits are however slightly lower than those of Ice Complex deposits, fluvial/alluvial deposits (Schirrmeister et al., 2011b), and Buor-Khaya Bay surface sediments (Vonk et al., 2012; Fig. 4). Average $\delta^{13}$C values of organic carbon at the thaw front of the three subsea permafrost cores (-24.9 ± 1.0‰, n = 153; Figs. 3 and 4) fell also at the upper end of this range. Total nitrogen averaged 0.08 ± 0.03% of dry weight (Table S3), and the mass ratio of organic carbon over total nitrogen (OC/TN) was 9.31 ± 2.48 across the subsea permafrost samples (Figs. 3 and 4). Taken together, the comparatively coarse grain sizes,

low organic carbon contents and enriched $\delta^{13}$C values of subsea permafrost at the current thaw front at the study site are consistent with an at least partly fluvial/alluvial deposition.

Subsea permafrost thaw on the ESAS might expose increasing amounts of organic matter to degradation. Although organic carbon content was overall low across the studied samples, the high rates of permafrost thaw of on average 14 cm year$^{-1}$ in the study area (Shakhova et al., 2017) correspond to the thaw-out of an additional 1.4 kg OC m$^{-2}$ year$^{-1}$, assuming 0.73% organic

carbon and 1.32 g dry weight cm$^{-3}$ bulk density (average of 2D-13 and 4D-12; data not shown). Understanding the molecular composition and microbial degradability of the increasing pool of thawed subsea permafrost organic carbon will be crucial for dissecting sources of CH$_4$ on the ESAS and constraining the Arctic greenhouse gas budget.



### 3.3 Plant biomarkers at the thaw front of subsea permafrost in the Buor-Khaya Bay

Ratios between lignin phenols reflect the relative contribution of organic matter from different vegetation types. The S/V and C/V ratios measured at the thaw front of subsea permafrost averaged $0.59 \pm 0.18$ and $0.34 \pm 0.25$, respectively, indicating a contribution of both woody gymnosperm and non-woody angiosperm tissues (Fig. 5b). Non-woody angiosperm vegetation is

characteristic for tundra landscapes with abundant shrubs or grasses that dominated Beringian landscapes in the late Pleistocene as well as today, and is reflected in high S/V and C/V ratios in Pleistocene Ice Complex deposits and Holocene active layer and permafrost deposits (Tesi et al., 2014; Winterfeld et al., 2015). In contrast, woody gymnosperm material in the studied subsea permafrost samples might not be autochthonous but transported by rivers. Vegetation reconstructions suggest that the tree line in the region has been south of the Buor-Khay Bay since the MIS 5e interglacial (Anderson and Lozhkin, 2001;

Kienast et al., 2011; Wetterich et al., 2009; Zimmermann et al., 2017), i.e., for the past 115 ka (Andreev et al., 2011). By comparison, OSL-dating indicates an age of 65-72 ka for the 3 m section at the thaw front of core 4D-12 that was analyzed for biomarkers, and younger ages for the corresponding sections of the other cores. A weak positive correlation of specific surface area of minerals with S/V ratios, and a stronger correlation with C/V ratios (Table 2) further indicate a higher contribution of woody gymnosperm tissues in coarse- than in fine-grained sediments. This also supports fluvial transport of

organic matter from boreal forests in the south. A similar mechanism can be observed under present conditions in eastern Siberia, where a high contribution of woody gymnosperm tissues in organic material suspended in the Lena River and deposited in surface sediments of the Buor-Khaya Bay has been inferred to reflect transport of organic matter from boreal forests in the south of the Lena catchment (Tesi et al., 2014; Winterfeld et al., 2015). However, subsea permafrost was characterized by lower values of the lignin degradation proxies Sd/Sl, Vd/Vl and 3,5-Bd/V (on average $0.33 \pm 0.14$, $0.48 \pm 0.15$, and $0.08 \pm$

$0.07$, respectively) than recent Lena River suspended material and Buor-Khaya Bay sediments (Tesi et al., 2014; Winterfeld et al., 2015; Fig. 6), suggesting a lower degree of lignin degradation under aerobic conditions and consequently shorter transport pathways.

Overall, the subsea permafrost samples studied here were characterized by low lignin contents, even considering the low organic carbon content. Lignin accounted for on average $5.31 \pm 5.21$ mg g$^{-1}$ OC in subsea permafrost (n = 56), a range

considerably lower than that of terrestrial deposits in eastern Siberia, recent Lena River suspended material, as well as recent Buor-Khaya Bay sediments (Fig. 6). The low lignin concentrations of subsea permafrost might reflect preferential lignin degradation after mobilization from terrestrial deposits, river transport and re-deposition. Incubation as well as field studies under anoxic conditions such as in subsea permafrost, however, rather suggest similar or even lower rates of lignin compared to bulk organic carbon degradation (Anwar et al., 2004; Chen et al., 2018; Dao et al., 2018; Dittmar and Lara, 2001; Gao et

al., 2016; Louchouarn et al., 1997; Opsahl and Benner, 1995). Alternatively, the low lignin concentrations might point at the dilution with organic carbon from other sources than higher land plants, for instance from moss-dominated peatlands. Lignin content per organic carbon was negatively correlated with P/V ratios (Table 2) that have been suggested as peat markers (Williams et al., 1998), possibly supporting a higher contribution of mosses to low-lignin samples.



### 3.4 Variability of subsea permafrost organic matter

Although this study represents a significant improvement of our understanding of subsea permafrost organic matter by describing organic carbon quantity and quality at the thaw front of three drill cores at high depth resolution (n = 153), our data are obviously not representative for the full depth (up to 600 m; Romanovskii et al., 2004) and spatial extent (up to $1.6 \times 10^6$

$km^2$; Fig. 1) of subsea permafrost on the ESAS. Even within the comparatively narrow range of our samples, we observed high variability in measured parameters. Statistical analysis suggests that the within- and between-core variability of organic matter at the thaw front of subsea permafrost reflected different sources, deposition regimes or degradation states at the time of deposition, rather than progressing decomposition upon recent thaw. Correlations between analyzed parameters show that coarse-grained material with low specific surface area was characterized by lower organic carbon and total nitrogen content

compared to fine-grained material (Table 2). The increase in organic carbon content with specific surface area was not proportional; the organic carbon load per surface area significantly decreased from coarse to fine material. Coarse-grained material further showed higher OC/TN ratios, and more enriched $\delta^{13}C$ values than fine-grained material, as well as lower S/V and C/V ratios and higher P/V ratios that indicate a lower contribution of non-woody angiosperm, and higher contribution of woody gymnosperm and moss tissues (see Sect. 3.3). By contrast, the lignin degradation proxies Sd/Sl, Vd/Vl, and 3,5-Bd/V

were weakly related to bulk organic matter properties. Positive correlations were only observed between Sd/Sl as well as 3,5-Bd/V and organic carbon normalized by specific surface area, as well as Vd/Vl and P/V. Negative correlations were observed between 3,5-Bd/V and specific surface area, as well as Vd/Vl and lignin content normalized by organic carbon. Taken together, these findings point at the accumulation of material of different origin at varying proportions over time.

High-resolution samples from above and below the IBPT represent a continuum from organic matter that has been frozen since

the Pleistocene, to organic matter that has thawed over the past decades. Linear interpolation between IBPT positions measured in 1982/3 and 2013 suggests the onset of thaw at the upper limit of the high-resolution sections studied in detail 10 years before drilling in core 4D-13 and 20 years before drilling in core 2D-13. Organic matter properties above and below the IBPT, as well as correlations with distance from the IBPT in the thawed part of the cores were examined to deduce any resolvable changes induced after thaw. There were few significant effects and those were non-systematic in direction (Table S5), and occurred

abruptly (Fig. 3), also supporting rather differences in organic matter sources than the imprint of gradually progressing decomposition upon thaw. Nevertheless, although organic matter degradation may alter several of the measured parameters (e.g., organic carbon content, OC/TN ratios, $\delta^{13}C$ values, lignin phenol ratios), none of these parameters is likely sensitive enough to capture the low rates of decomposition expected under the cold and anoxic conditions at the thaw front of subsea permafrost.

Among the three cores, 4D-13 showed the highest variability in organic carbon and total nitrogen content, $\delta^{13}C$ values, lignin concentrations, as well as C/V, Sd/Sl, Vd/Vl, and 3,5-Bd/V ratios (Fig. 3; Tables S3 and S4). This variability contrasts with the rather similar grain size distributions along the entire core, including the 3 m section at the IBPT that was analyzed in detail for organic matter properties (Fig. 2). Core 4D-13 was drilled closest to Muostakh Island, had the shallowest IBPT (Fig. 2)

and likely the youngest organic matter at the IBPT that was analyzed in detail. This coincidence might hint at a trend towards more homogeneous organic matter properties with age.

## 4 Conclusions

Subsea permafrost on the Arctic Ocean shelves might be rapidly thawing due to natural and anthropogenic warming, highlighting the need to constrain the quantity and quality of organic carbon currently stored in subsea permafrost. Based on three drill cores of 21-56 m length from the Buor-Khay Bay on the ESAS, we show high spatial and temporal variability of lithological properties that reflect sediment deposition in a heterogeneous and dynamic landscape over the past 160 ka. Organic matter at the current thaw front in this region was characterized by low organic carbon contents and enriched $\delta^{13}C$ values similar to the ranges previously observed in Pleistocene fluvial/alluvial deposits at nearby terrestrial sites, and was derived from both tundra and boreal forest vegetation, at least the latter most likely transported by rivers. In spite of the low organic carbon content, the high rates of permafrost thaw observed at the study site of 14 cm year$^{-1}$ correspond to a thaw-out of 1.6 kg OC m$^{-2}$ year$^{-1}$. Constraining the susceptibility of this vast and heterogeneous organic carbon pool to degradation upon thaw is urgently needed for improving estimates of greenhouse gas emissions from permafrost and its feedback to climate warming.

## 5 Data availability

Data presented in this study are available in the Supplementary Material (Tables S1-S4).

## 6 Author contribution

BW, IS and ÖG designed the study; OD, AR, DK, VT as well as IS conducted the drilling campaigns and processed cores and samples. HA and MJ provided OSL data. Analysis of bulk organic matter properties was performed by TT, of lignin phenols by BW, and of specific surface area by HJ. All authors contributed to writing of the manuscript, led by BW and ÖG.

## 7 Competing interests

The authors declare that they have no conflict of interest.

## 8 Acknowledgements

This study was funded by the Swedish Research Council (VR grant numbers 621-2013-5297 and 2017-01601 to ÖG), the European Research Council (ERC-AdG CC-TOP, grant number 695331 to ÖG), the Russian Science Foundation (grant



number 15-17-20032 to NS), and the Russian Government (grant number 14, Z50.31.0012 to IS). We further thank Rajendra
Shrestha (Lund University) for OSL sample preparation.

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



Table 1. Overview of the subsea permafrost drill cores 4D-13, 2D-13, 4D-12 from the Buor-Khaya Bay. Time since inundation, IBPT depth at the time of sampling 2012/13, as well as rate of IBPT deepening between 1982/83 and 2013 are from Shakhova et al. (2017). All depths are in m below the sea floor.

| | 4D-13 | 2D-13 | 4D-12 |
|---|---|---|---|
| Coordinates | 71°37'03''N, 129°55'19''E | 71°37'44''N, 129°52'53''E | 71°37'46''N, 129°52'32''E |
| Distance from Muostakh Isl. | 600 m | 2500 m | 2900 m |
| Time since inundation | 145 years | 460 years | 511 years |
| Water depth | 2.5 m | 3.4 m | 2.5 m |
| Core length | 21.0 m | 30.4 m | 55.7 m |
| IBPT depth 2012/13 | 8.9 m | 15.9 m | 23.9 m |
| IBPT deepening | $18.3 \pm 0.1$ cm year$^{-1}$ | $9.3 \pm 1.7$ cm year$^{-1}$ | n.a. |



Table 2. Correlations between individual parameters measured at the thaw front of three subsea permafrost drill cores from the Buor-Khaya Bay, including specific surface area of minerals (SSA), organic carbon (OC) content normalized by dry weight (d.w.) and SSA, total nitrogen (TN), OC/TN ratios, $\delta^{13}C$ values, lignin concentrations, as well as biomarker ratios (S/V, C/V, P/V, Sd/Sl, Vd/Vl, 3,5-Bd/V). Presented are Spearman's correlation coefficients for correlations significant at $p < 0.05$ (n.s., not significant).

| | SSA ($m^2$ $g^{-1}$ d.w.) | OC (%) | OC (mg $m^{-2}$) | TN (%) | OC/TN | $\delta^{13}C$ (‰) | Lignin (mg $g^{-1}$ OC) | S/V | C/V | P/V | Sd/Sl | Vd/Vl |
|---|---|---|---|---|---|---|---|---|---|---|---|---|
| OC (%) | +0.337 | | | | | | | | | | | |
| TN (%) | +0.651 | +0.589 | | | | | | | | | | |
| OC (mg $m^{-2}$) | −0.902 | n.s. | | −0.394 | | | | | | | | |
| OC/TN | −0.580 | n.s. | +0.653 | −0.714 | | | | | | | | |
| $\delta^{13}C$ | −0.724 | −0.501 | +0.523 | −0.532 | +0.294 | | | | | | | |
| Lignin (mg $g^{-1}$ OC) | n.s. | +0.456 | n.s. | +0.406 | n.s. | −0.431 | | | | | | |
| S/V | +0.283 | +0.299 | n.s. | +0.500 | −0.310 | n.s. | +0.599 | | | | | |
| C/V | +0.547 | +0.573 | −0.323 | +0.781 | −0.461 | −0.558 | +0.451 | +0.568 | | | | |
| P/V | −0.278 | n.s. | +0.311 | n.s. | n.s. | n.s. | −0.482 | n.s. | n.s. | | | |
| Sd/Sl | n.s. | n.s. | +0.278 | n.s. | n.s. | n.s. | n.s. | n.s. | n.s. | n.s. | | |
| Vd/Vl | n.s. | n.s. | n.s. | n.s. | n.s. | n.s. | −0.364 | n.s. | n.s. | +0.277 | +0.649 | |
| 3,5-Bd/V | −0.368 | n.s. | +0.406 | n.s. | n.s. | n.s. | n.s. | n.s. | n.s. | n.s. | n.s. | +0.282 |





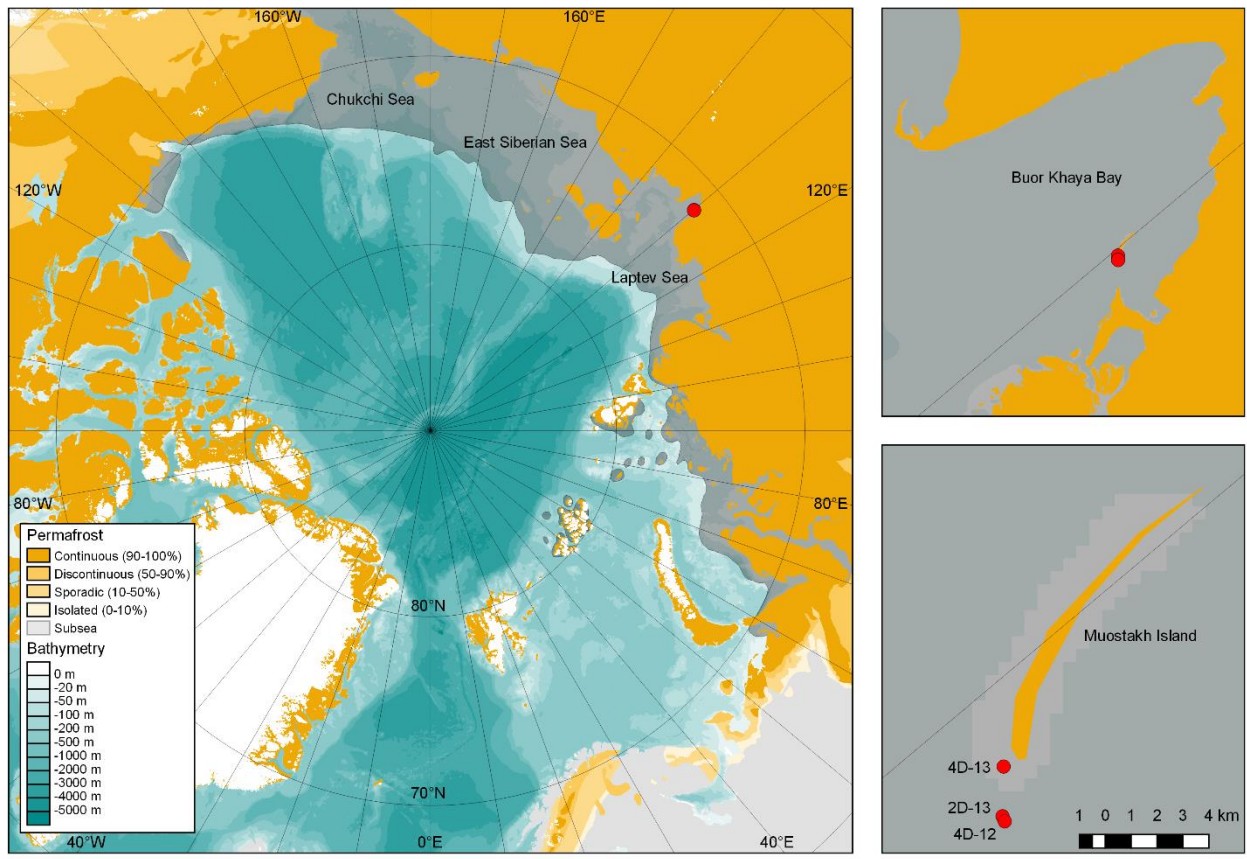

**Figure 1: Extent of terrestrial and subsea permafrost across the Arctic (Brown et al., 2001). Smaller panels show the study region focusing on the Buor-Khaya Bay and Muostakh Island; red dots indicate the drilling sites of the 4D-13, 2D-13, and 4D-12 subsea permafrost cores. The Arctic Ocean bathymetric map is based on IBCAO v3 (Jakobsson et al., 2012).**





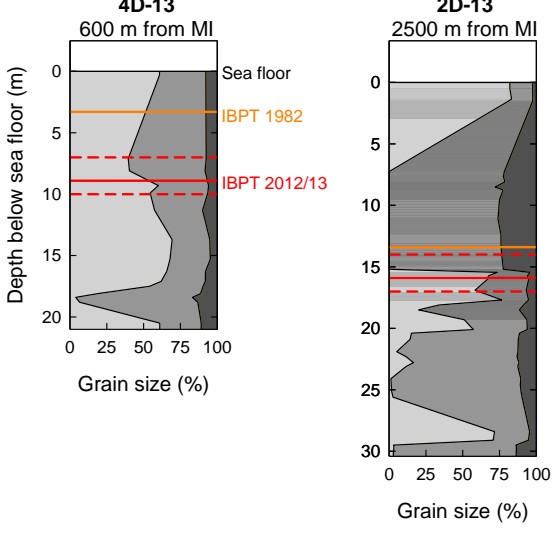

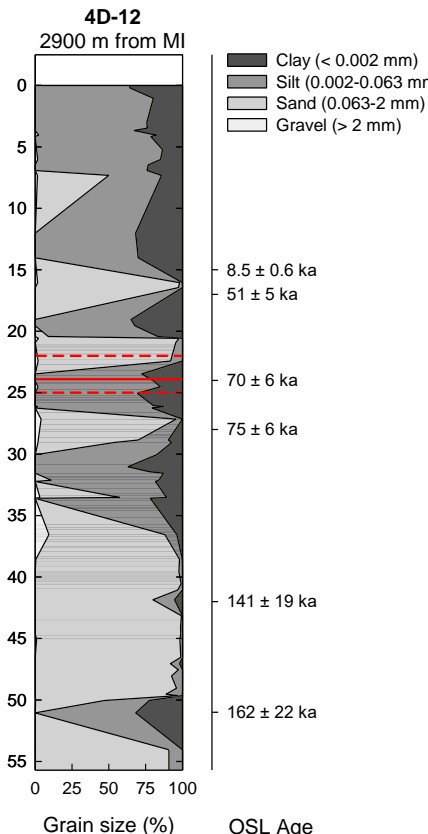

**Figure 2: Grain size distribution of the three subsea permafrost cores 4D-13, 2D-13, and 4D-12, drilled 600 m, 2500 m, and 2900 m from Muostakh Island in the Buor-Khaya Bay, as well as OSL ages for the core 4D-12. The IBPT position at the time of drilling in 2012/13 as well as in 1982 is indicated by red and orange lines, respectively. The IBPT position of core 4D-12 has not been determined**
5 **in 1982. The interval marked between the dashed red lines shows the 3 m sections where detailed organic matter analyses have been performed. Grain size and IBPT data are from Shakhova et al. (2017); detailed grain size data are provided in Table S2.**



**Figure 3: Depth profiles of the subsea permafrost cores 4D-13, 2D-13, and 4D-12, for 3 m sections at the IBPT (indicated by the red line). Plotted are specific surface area (SSA), organic carbon content (OC), OC/TN ratios and δ¹³C values, as well as lignin phenol concentrations, lignin source proxies (S/V, C/V, P/V) and lignin degradation proxies (Sd/Sl, Vd/Vl, 3,5-Bd/V). Ages based on linear interpolation of OSL-dates are shown for core 4D-12.**





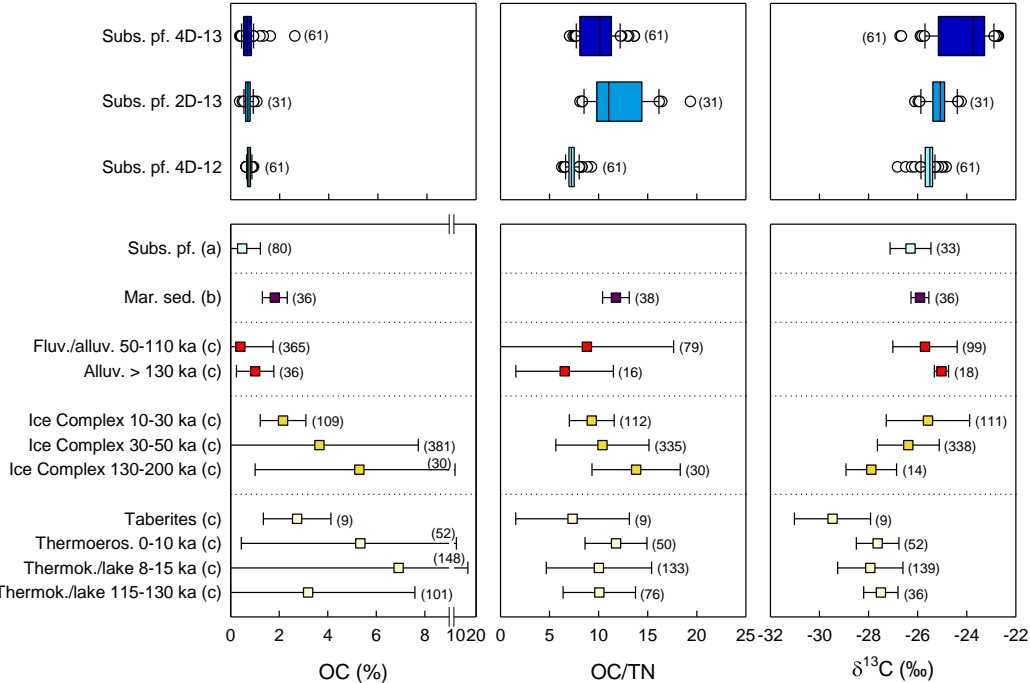

**Figure 4: Comparison of organic carbon (OC) content, OC/TN ratios and $\delta^{13}$C values at the thaw front of the subsea permafrost cores 4D-13, 2D-13, and 4D-12 with other deposits in the region. Subsea permafrost data from this study are presented as box plots of medians with 25th and 75th percentiles as box limits, 10th and 90th percentiles as whiskers, and all outliers indicated. Data for an additional subsea permafrost core and marine surface sediments of the Buor-Khaya Bay, as well as fluvial-alluvial deposits, Ice Complex deposits, taberites, thermoerosion, thermokarst, and lake deposits of different deposition age from eastern Siberia are from previous publications (a, Overduin et al., 2015a, 2015b; b, Vonk et al., 2012; c, Schirrmeister et al., 2011b; alignment of stratigraphy and age following Andreev et al., 2011), and presented as means with standard deviations as error bars. Numbers of observations are in brackets.**





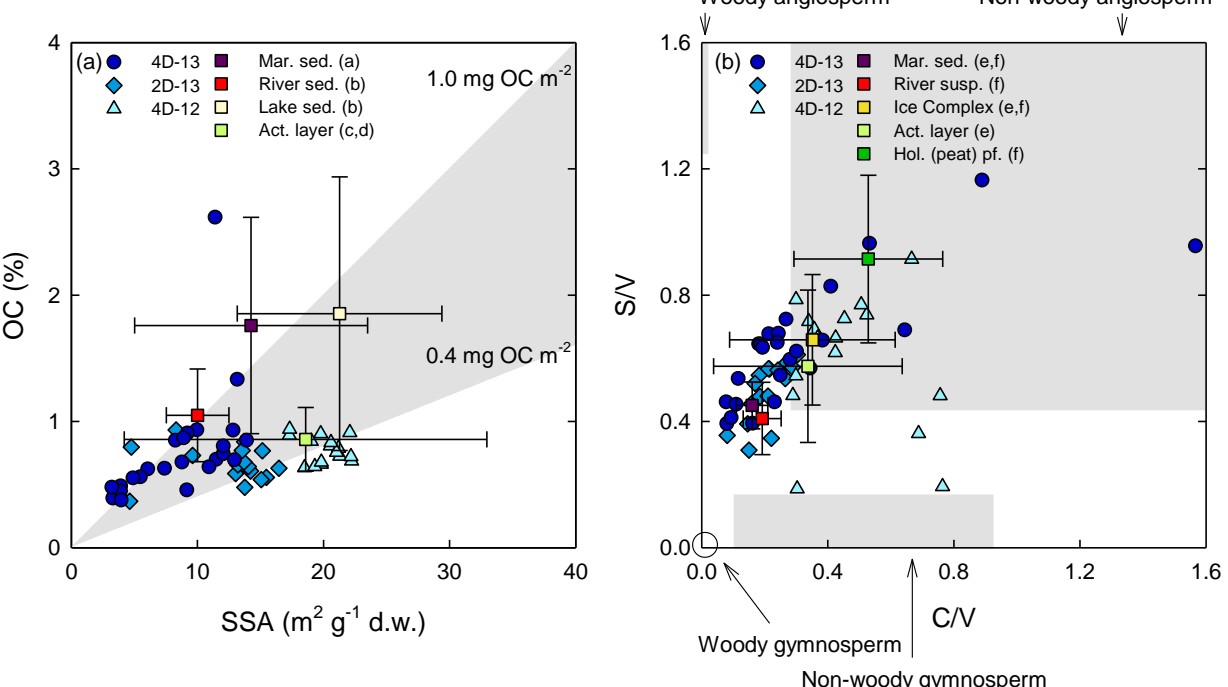

**Figure 5: (a) Specific surface area (SSA) vs organic carbon (OC) content for 3 m sections at the thaw front of the subsea permafrost drill cores 4D-13, 2D-13, and 4D-12. The grey area indicates the range between surface area normalized organic carbon contents of 0.4 and 1.0 mg OC m$^{-2}$ that is typically observed in sediments (Blair and Aller, 2012). Additional data from previous studies on**

5   **recent sediments of the Buor-Khaya Bay (a, Karlsson et al., 2015), the Mackenzie river and adjacent lakes (b, Vonk et al., 2015), as well as mineral active layer soils from Western Siberia (c, Buchkina et al., 1998) and Spitsbergen (d, Cieśla et al., 2018) are shown as means with standard deviations. (b) Biplot of S/V over C/V lignin phenol ratios for the same samples. The grey areas indicate the typical S/V and C/V ranges of woody angiosperm, non-woody angiosperm, woody gymnosperm and non-woody gymnosperm tissues (Dao et al., 2018; Goñi and Hedges, 1992; Hedges and Mann, 1979; Hedges and Parker, 1976; Otto and Simpson, 2006; Winterfeld**

10   **et al., 2015). Additional data on marine surface sediments from the Buor-Khaya Bay, Lena River suspended matter, Ice Complex deposits, active layer and Holocene permafrost samples from eastern Siberia are shown as means with standard deviations (e, Tesi et al., 2014; f, Winterfeld et al., 2015).**





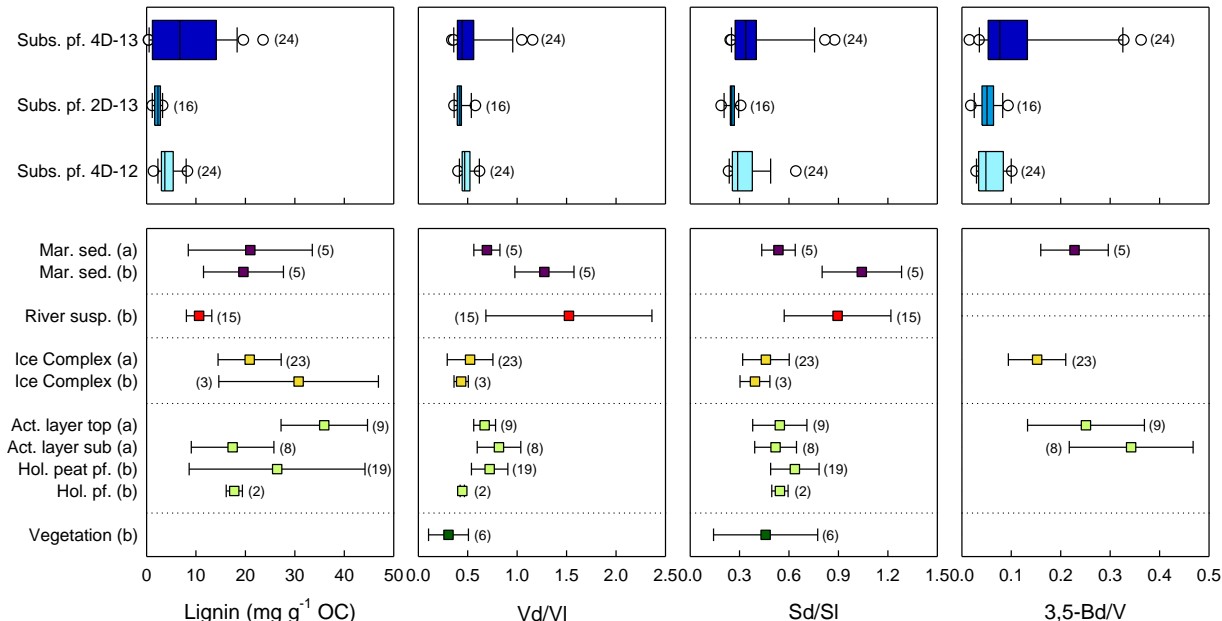

**Figure 6: Comparison of lignin content and lignin degradation proxies (Sd/Sl, Vd/Vl, 3,5-Bd/V) at the thaw front of the subsea permafrost cores 4D-13, 2D-13, and 4D-12 with other deposits in the region. Subsea permafrost data from this study are presented as box plots of medians with 25th and 75th percentiles as box limits, 10th and 90th percentiles as whiskers, and all outliers indicated.**
5 **Data for marine surface sediments of the Buor-Khaya Bay, Lena River suspended matter, as well as Ice Complex deposits, active layer top- and subsoil, Holocene permafrost and tundra vegetation from eastern Siberia are from previous publications (a, Tesi et al., 2014; b, Winterfeld et al., 2015) and presented as means with standard deviations as error bars. Numbers of observations are in brackets.**