# Peer review of "Organic matter across subsea permafrost thaw horizons on the East Siberian Arctic Shelf"

_The Cryosphere, 2018_

## Referee Comment (RC1) · Anonymous Referee #1 · 21 Jan 2019

General comments 1) I am puzzled by this manuscript. The authors provide a large data set and the manuscript is well written. I do not have a lot of criticism about each chapter itself, however, I found myself asking what the real aim of the study was since each of the chapters seems to navigate towards different topics that do not have a common scope. I think this is mostly due to the fact that there are no distinct changes between proxy data above and below the IBPT, which was likely not expected. Accordingly, other aspects are discussed, but I am missing a little more focus on how this helps understanding subsea permafrost thaw (feedbacks). Here is my summary of the main points of each chapter, which hopefully shows why I think the overall manuscript needs a more structured aim. The introduction provides background on the ESAS and potential CH4 release from subsea permafrost due to climate change and per-

mafrost thaw. This permafrost carbon feedback and CH4 dynamics particularly is/are not mentioned/discussed again. If this will be picked up again, the discussion should acknowledge that looking only at lignin phenols and no other compound classes provides a limited assessment of the molecular composition a very limited view on microbial degradability. The discussion starts (chapter 3.1) with a comparison of the grain size data with other Siberian permafrost deposits in order to constrain the origin of these deposits. This certainly is important background knowledge, but the data are acknowledged to having been published by Shakova et al. (2017) already and cover a much wider depth range than what is relevant for the remainder of the discussion and understanding IBPT deepening and feedbacks. This chapter could be significantly shortened and mostly reference Shakova et al. (2017) to provide the necessary background information. In chapter 3.2, the bulk characteristics are discussed in comparison to other high latitude regions including the ESAS as well as Alaska and Svalbard (for OC loading) and between different deposit types (marine and lacustrine sediments, soils). This broader geographical context is not discussed in any of the other chapters and I wonder how OC loadings in Svalbard active layer soils (Svalbard itself is a very different system) help to understand changes associated with subsea permafrost thaw in the three investigated cores? Chapter 3.3 provides constraints on lignin phenol sources/origin and an assessment of the degradation state of these lignin phenols both in comparison with other studies in the Buor Khaya vicinity. To tie it to the previous chapter, how does that compare to other high latitude settings? Also, how much do we learn from lignin phenol degradation state alone without additional information from other (less refractory) compound classes? Chapter 3.4 finally provides the discussion one would expect – the comparison of proxy data above and below the IBPT. This chapter is relatively brief and while the statistical analysis does not show significant changes between frozen and thawed subsea permafrost in these three cores, there is some variability that could be discussed in a little more detail (than the three sentences at the end of the chapter). Also, I would expect an assessment of whether significant changes across the IBPT can be expected in the first place given that the deposits are

so heterogenous? Are 30 years of thaw enough to expect a significant change in bulk OC characteristics and lignin phenol abundances (even in a homogenous deposit)?

2) I am not quite convinced by the argument made regarding the IBPT depths and sediment age in cores 2D-13 and 4D-13. The jump from 51ka at 17m depth to 8.5ka at 15m in core 4D-12 is explained as either a period of low deposition or an erosional event (page 6). The IBPT in core 2D-13 is at ca. 16m (the "peak jump" in AD-12), the IBPT in core 4D-13 at about 9m. Neither the lithological nor the organic proxies allow for any easy core correlation. This is likely due to the different depositional settings and indicates that the depositional history is different, thus, also invoking a period of low deposition or and erosional event and expecting younger ages for cores 2D-13 and 4D-13 is not unequivocally justified. These two cores may not be affected by the same processes and sedimentation rates might be very different. Please provide additional age constraints (OSL or maybe 14C?) or discuss this much more carefully.

3) you may want to consider additional lignin phenol data from the Buor Khaya Bay: Ulyantseva et al. (2018) The Molecular Composition of Lignin as an Indicator of Sub-aqueous Permafrost Thawing. doi: 10.1134/S1028334X1810029X.

Specific comments Page 2 l.3-5: it would read nicer to combine the subordinate clause with the remainder of the sentence. l.6: "destroyed by erosion" or eroded? l.7: "This process" - which process are you referring to? Erosion or inundation? l.8: add super-script to unit. l.10: "due to the changing thermal gradient"?

Page 3 l.7: capitalize "delta". l.12: please define the origin of Yedoma (vs. fluvial and alluvial deposits). l.20: change header or include additional header after l.26. Most of the chapter does not reference the field work, but either IBPT deepening rates or laboratory methodology/sample processing.

Page 4 l.11-12: move last sentence to introduction or discussion.

Page 5 l.11: is OC not total OC? The distinction is made for N, but not OC. l.30-31:

please use common italics notation for phenols.

Page 6 l.1-9: please use common italics notation for phenols. l.5: add information that all phenol concentrations were normalized to g OC.

Page 7 l.1-3: what is the threshold value to differentiate unimodal and bimodal distributions? Based on Fig. 2, 4D-12 also has substantial clay contribution (some 25% or so) in those intervals dominated by silt, but this does not qualify as a bimodal distribution? l.9-27: it would be nice if the argument order in this paragraph was reversed, starting e.g. with the sentence in l.20 so the connection to the above paragraph is more obvious. l.19-20: does TC allow citing articles in review? l.22-23: more similar in comparison to?

Page 8 l.8: change to "normalized to" l.15: what can we learn from comparison with active layer soils in Svalbard? That is a very different Arctic setting. l.18: O'Leary (1988) does not provide $\delta13C$ values for plants in East Siberia. l.18-21: please provide endmember values for the cited high latitude references. l.30-32: you are only looking at lignin phenols (not the molecular composition), so the data provide a very limited view on microbial degradability, since no other compound classes are assessed and lignin is a very refractory material to start with.

Page 9 l.18-22: while the acid to aldehyde ratio is used to determine the degree of aerobic decomposition, isn't the fact subsea permafrost in this area is anoxic below the SMTZ (which coincides with the IBPT; e.g. Winkel et al. 2018, Scientific Reports, doi:10.1038/s41598-018-19505-9) suggesting that this is rather a function of age/exposure time to aerobic conditions? Irrespective of transport distance and duration, if the OC from the boreal forests in the South was stored in soils prior to erosion and export, it is likely much more degraded. l.26-27: this statement contradicts with the previous statement arguing for a lower degree of lignin degradation in subsea permafrost samples vs. riverine and surface sediments (l.18-22). Also, based on Fig. 6, the heterogeneity in terrestrial deposits and surface sediments is large and within SD

at least agrees with the concentrations in core 4D-13.

Page 10 l.2-3: I would argue that the data set is too limited to assess OC qualities. It allows to assess lignin quality; for OC quality, a comprehensive data set including various compound classes would be needed. l.8-14: this entire paragraph is phrased as if these measurements were performed on actual density fractions, which is not the case. This should be re-phrased to acknowledge that "samples with finer or coarser grain size distributions. . ."

Fig. 1 Please add inset boxes and panel IDs and increase the font size in all maps and the legend. Add contour lines and legends to the small maps on the right. These also miss coordinate systems.

Fig. 2 What exactly is the reference point for the distances? The coastline? l. 4: change "has not been" to "was not"

Fig. 4 Please include abbreviations in figure caption or use unabbreviated labels. There are many more endmember values available for %OC and OC/TN, why are they not included? One example, several marine sediments are referenced in chapter 3.2, which could be added to the plot. Why are the own data shown as box-whisker plots, but the reference data are not? Comparing medians and means is not straightforward.

Fig. 6 Please include abbreviations in figure caption or use unabbreviated labels.

---

## Referee Comment (RC2) · Serov (Referee) · 23 Jan 2019

The manuscript describes content and composition of organic matter within a recently thawed permafrost horizon encountered in 3 re-drilled offshore boreholes in the Laptev Sea. The results indicate that the bulk organic matter content is rather low, which resonates with previously published works. Lignin phenol concentrations are unexpectedly low, which the authors suggest is due to its preferential degradation, river transportation and re-deposition. Using grain-size distribution, the authors attempt to reconstruct depositional environment, which would support their interpretations of the sources of the organic matter. Interpretation of analysis points toward rather heterogeneous content, origin and composition of organic matter within a small study area (maximum distance between the boreholes seems to be ~2 km) overriding any detectable effects

of organic matter degradation.

The data quality and analytical techniques used are beyond dispute. The text is clearly written, the very general title adequately reflects somewhat diluted focus of the paper, and the subject matter of the manuscript may fit the scopes of The Cryosphere journal. However, I found a few rather large-scale issues that prevent me from recommending this manuscript for publication at its current state.

1. Lack of focus. It is not clear what this paper is about. When reading the Abstract and, partly, introduction I was sure it would tell a story of a recent thaw leading to organic matter degradation and greenhouse gas production. These first parts of text unambiguously point towards that. Yet, the results subchapters and discussion talk about depositional settings, permafrost evolution throughout the last 160 ka, origin and transportation of organic matter, etc. What all these findings have to do with the truly unique observations of rapid and recent thaw? In fact, investigating any permafrost core in the region (no matter if it is frozen or thawed, first time drilled or re-drilled) would reveal the same facts on sediment deposition, organic carbon contents, and nature of organic matter. In addition, I have to mention that the paper by Shakhova et al., 2017 covered substantial part of what is discussed in the manuscript. 2. Mismatch of the declared research question(s) and results-discussion. The manuscript stresses on importance of permafrost thaw for climate gas budgets. Therefore, degradation state of organic matter in the cores would be the most relevant part of the manuscript. However, on the second-to last page, the manuscript says that none of the analyzed parameters (org. carbon content, d13C, lignin phenol ratios, OC/TN ) is sensitive enough to capture the decomposition of organics matter. The other findings do not seem to have direct relation to the climate gas problem, or the manuscript does not explicitly show such connections. Perhaps, one way to approach answering the question of CO2 and CH4 release from the thawed strata would be to calculate potential maximum and minimum scenarios of greenhouse gas generation in these particular settings and assuming the estimated thaw rates. Knowing that 1.6 kg of OC m-2 thaw-out every year does not
bring us to understanding the magnitude of associated climate gas production. 3. Watery conclusions. Throughout the manuscript, I noticed that several lines of evidence lead to somewhat empty conclusions, such as Âńtaken together these findings point at the accumulation of material of different origin at varying proportions..", or the ones saying that more research is urgently needed. I suggest cutting off all parts of the manuscript not leading to new/important results. 4. The above-mentioned issues brought me to a conclusion that I cannot say that the paper advances our understanding of how important the extremely fast permafrost thaw is for modulating CO2 and CH4 release. The manuscript does thoroughly describe several properties of the organic matter within investigated cores and provides a discussion of the sediment deposition and thaw history (which overlaps with the previously published paper by Shakhova et al., 2017). This does contribute to a growing number of site-specific descriptive studies of OM composition within permafrost, but does not provide a transformative step towards understanding the consequences of rapid permafrost thaw.

Please, find the smaller scale issues and some technical corrections in the enclosed .pdf file.

Sincerely, Pavel Serov

Please also note the supplement to this comment:
https://www.the-cryosphere-discuss.net/tc-2018-229/tc-2018-229-RC2-supplement.pdf

**Supplement:**

[revised manuscript text omitted]

---

## Editor Comment (EC1) · Dominé (Editor) · 24 Jan 2019

Dear Authors,

Both reviewers conclude that your article is nicely written and contains nice and potentially useful data, but point to a major flaw in your paper: a lack of scope and it is not clear what question this study answers. There is also a significant overlap with an earlier paper by Shakova et al. (2017). These are major issue that could lead to rejection at this stage. However, if you feel you can produce a very significantly revised version, I leave this option open to you. Should you choose to do so, I will be very attentive to the following points:

1- A clear and logical line of thought must be followed, with clear questions posed at

the start and a strategy that logically answers that question, at least partially, while possibly stressing outstanding issues.

2- The manuscript must be significantly streamlined to avoid duplication of earlier work and above all to avoid superfluous digressions from the main line of thought.

Alternatively, you could decide to resubmit a new version of your paper as a new manuscript, which of course would take into account the reviewers' very helpful and constructive comments. I would be grateful if you would please let me know of your intentions within 2 weeks.

Best regards,

Florent Domine, editor.

―――――――――――――――――

---

## Author Comment (AC1) · 21 Feb 2019

Author responses and resulting edits to tc-2018-229

**Organic matter across subsea permafrost thaw horizons on the East Siberian Arctic Shelf**

by Birgit Wild, Natalia Shakhova, Oleg Dudarev, Alexey Ruban, Denis Kosmach, Vladimir Tumskoy, Tommaso Tesi, Hanna Joß, Helena Alexanderson, Martin Jakobsson, Alexey Mazurov, Igor Semiletov, Örjan Gustafsson

We thank both reviewers for their constructive comments. We recognize that we had not clearly communicated the motivation and scope of our study and have therefore substantially revised the manuscript to clarify such important context. The title has been adjusted to avoid misunderstandings, to "*Organic matter at the thaw front of subsea permafrost on the East Siberian Arctic Shelf*".

**Our study is motivated by the need for observational data on subsea permafrost**. Subsea permafrost represents a potentially large and vulnerable organic carbon pool that might be or become a relevant source of greenhouse gases to the atmosphere. In contrast to terrestrial permafrost where organic carbon stocks and greenhouse gas emissions are increasingly well constrained, carbon stocks and dynamics of subsea permafrost remain highly uncertain due to a paucity of observational data from these hard-to-access systems. Current models of permafrost carbon dynamics consequently have no choice but to either not consider subsea permafrost (e.g., Schuur et al., 2015) or work with large and uninformed assumptions (e.g., Lindgren et al., 2018).

Given the need to improve the understanding of subsea permafrost carbon dynamics, **this study compares organic matter at the current subsea permafrost thaw front with terrestrial permafrost deposits** that could serve as analogue systems. This would permit future studies to make more informed assumptions about subsea permafrost properties and responses, based on easily accessible and consequently better studied terrestrial permafrost deposits.

This overarching aim of the study is specified in three objectives using a set of subsea drill cores, to

(1) **determine the time of subsea permafrost deposition** focusing on the current thaw front; the optically stimulated luminescence dating presented here represents the, to our knowledge, first dating of a subsea permafrost core and facilitates comparison with terrestrial permafrost deposited over similar time frames;

(2) **describe organic matter at the subsea permafrost thaw front**, quantitatively and qualitatively based on organic carbon content and isotopic composition as well as biomarker data; and

(3) **compare organic matter at the subsea permafrost thaw front with terrestrial permafrost** deposited under varying environmental conditions in the study region.

By providing observational data on subsea permafrost organic matter, our study represents an important step in advancing our understanding of this potentially large and vulnerable pool, moving from data-lean or even data-void large assumptions to solid data and thus scientifically more credible estimates.

We now further emphasize that while our study builds on the same subsea permafrost drill cores as a previous study (Shakhova et al., 2017) and consequently shares a description of basic data, it is quite novel and complementary in its focus on organic matter; we expect many studies focusing on a variety of aspects of subsea permafrost to come, based on this unique set of hard-to-retrieve subsea drill core samples.

All reviewer comments are addressed in detail in the following. First, reviewer comments are reproduced in blue italics, immediately followed by the corresponding author responses in normal black font.

**Anonymous Referee #1**

*General comments*

*1) I am puzzled by this manuscript. The authors provide a large data set and the manuscript is well written. I do not have a lot of criticism about each chapter itself, however, I found myself asking what the real aim of the study was since each of the chapters seems to navigate towards different topics that do not have a common scope. I think this is mostly due to the fact that there are no distinct changes between proxy data above and below the IBPT, which was likely not expected. Accordingly, other aspects are discussed, but I am missing a little more focus on how this helps understanding subsea permafrost thaw (feedbacks). Here is my summary of the main points of each chapter, which hopefully shows why I think the overall manuscript needs a more structured aim. The introduction provides background on the ESAS and potential CH4 release from subsea permafrost due to climate change and permafrost thaw. This permafrost carbon feedback and CH4 dynamics particularly is/are not mentioned/discussed again. If this will be picked up again, the discussion should acknowledge that looking only at lignin phenols and no other compound classes provides a limited assessment of the molecular composition a very limited view on microbial degradability. The discussion starts (chapter 3.1) with a comparison of the grain size data with other Siberian permafrost deposits in order to constrain the origin of these deposits. This certainly is important background knowledge, but the data are acknowledged to having been published by Shakova et al. (2017) already and cover a much wider depth range than what is relevant for the remainder of the discussion and understanding IBPT deepening and feedbacks. This chapter could be significantly shortened and mostly reference Shakova et al. (2017) to provide the necessary background information. In chapter 3.2, the bulk characteristics are discussed in comparison to other high latitude regions including the ESAS as well as Alaska and Svalbard (for OC loading) and between different deposit types (marine and lacustrine sediments, soils). This broader geographical context is not discussed in any of the other chapters and I wonder how OC loadings in Svalbard active layer soils (Svalbard itself is a very different system) help to understand changes associated with subsea permafrost thaw in the three investigated cores? Chapter 3.3 provides constraints on lignin phenol sources/origin and an assessment of the degradation state of these lignin phenols both in comparison with other studies in the Buor Khaya vicinity. To tie it to the previous chapter, how does that compare to other high latitude settings? Also, how much do we learn from lignin phenol degradation state alone without additional information from other (less refractory) compound classes? Chapter 3.4 finally provides the discussion one would expect – the comparison of proxy data above and below the IBPT. This chapter is relatively brief and while the statistical analysis does not show significant changes between frozen and thawed subsea permafrost in these three cores, there is some variability that could be discussed in a little more detail (than the three sentences at the end of the chapter). Also, I would expect an assessment of whether significant changes across the IBPT can be expected in the first place given that the deposits are so heterogenous? Are 30 years of thaw enough to expect a significant change in bulk OC characteristics and lignin phenol abundances (even in a homogenous deposit)?*

Response: We thank the reviewer for the critical feedback that have helped to considerably improve the manuscript. In the following, we address the points brought up individually and explain the resulting changes to the manuscript.

First, we agree that we have not been sufficiently clear in explaining the focus of this study. While constraining the potential of subsea permafrost as a source of $CH_4$ to the atmosphere is a long-term aim and motivation for this line of research, $CH_4$ release is not the scope of the current study which takes some first steps at characterizing the organic matter in subsea permafrost. We had no expectations about changes in the studied parameters across the thaw front; the understanding of subsea permafrost organic matter and its degradation so far is in fact too limited to support hypotheses suggesting changes of individual parameters or a lack thereof within a few decades after thaw. It is a key aim of this study to shed some light into organic matter properties and dynamics at the subsea permafrost thaw front – a pre-requisite for further targeted research on these topics in the future. Based on the feedback by both reviewers and the editor, we have substantially revised the manuscript to better outline the motivation, scope and objectives and also adjusted the title; see also page 1 of this response letter for details.

Second, it is correct that grain size data have been previously published (Shakhova et al., 2017) and we cite this publication accordingly throughout the manuscript and in the corresponding figure legend. We also agree that these data cover a wider depth range than most of the Discussion that focuses on the current thaw front. Nevertheless, the interpretation of grain size data presented here goes well beyond that in Shakhova et al. as these data support the new optically stimulated luminescence dates to together place the subsea permafrost core in the stratigraphic context of the study region, and complement the comparison of subsea and terrestrial deposits to also consider lithological properties. Hence, the current manuscript uses the grain size data toward a very different objective than the earlier study. We consequently see value in presenting these data over a wider depth range to place the studied depth interval into context. We now clarify these two motivations in the Introduction and Results/Discussion sections.

Third, we agree that we have not explained well the comparison of subsea permafrost with other deposits. Subsea permafrost on the East Siberian Arctic Shelf developed when rising sea levels submerged terrestrial permafrost landscapes with intact and thermokarst-affected Ice Complex and other permafrost deposits, as well as lakes and rivers. It is not clear to which extent these permafrost deposits are preserved under ESAS waters and how the stratigraphy of subsea permafrost and coastal terrestrial permafrost deposits align. To contribute to bridging this gap, this study compares subsea permafrost material to a set of other deposits formed during the Pleistocene and preserved at terrestrial sites: (1) Pleistocene Ice Complex deposits, (2) Pleistocene fluvial/alluvial deposits and (3) thermokarst deposits developed during and at the end of the Pleistocene. This list is complemented by a set of Holocene deposits that are more abundant, better studied and thereby provide additional information and systems understanding: (4) Holocene permafrost deposits, (5) Holocene active layers, (6) suspended material from the Lena river that transports material derived from terrestrial permafrost to the Arctic Ocean, as well as (7) marine surface sediments from the Buor-Khaya Bay that receive input from the Lena river. Following the suggestion of Reviewer 1 below, a wider range of marine surface sediments has been added to the comparison to highlight differences in river influence and terrestrial carbon deposition. In response to the comment of Reviewer 1 above, the comparison has been further restricted to deposits from Siberia; we agree that the situation in particular on Svalbard is too different to justify an inclusion in this manuscript. The Abstract and Introduction have been extended to explain our motivation for comparing subsea and terrestrial permafrost deposits (see also above), and the description of the study area in the Material and Methods section as well as in Results/Discussion to define the geographic focus of this study and clarify the choice of deposits for comparison.

Finally, we also agree that we have not communicated our reasons for choosing lignin as a biomarker in this study. Lignin is frequently used in studies of sediment cores since degradation under anoxic conditions is slow and ratios between individual lignin phenols remain constant (e.g., Hamilton and Hedges, 1988; Louchouarn et al., 1997). Independence of lignin phenol ratios to decomposition after subsea permafrost thaw is a requirement for reliable comparison with still-frozen terrestrial permafrost; this assumption is supported by statistical analysis of differences in bulk and biomarker properties across the ice-bonded permafrost table (IBPT) that does not show systematic changes after thaw but rather suggests changes in organic matter input as the source of the observed variability. The source and degradation state proxies used here are consequently robust indicators of the state of organic matter at the time of deposition and unlikely to be substantially affected by degradation after subsea permafrost thaw; identifying differences in measured properties across the IBPT was thus not the primary focus of this manuscript. We have updated the manuscript Introduction to clarify the context and motivation for comparing properties across the IBPT (see also above). Furthermore, lignin data have also been reported for different permafrost deposits in the study region by previous studies (Tesi et al., 2014; Winterfeld et al., 2015), enabling the comparison of subsea and terrestrial permafrost deposits. Nevertheless, we agree that lignin is not representative for the diversity of chemical compounds that form organic matter. In the updated manuscript, we address the motivation for choosing lignin as a biomarker, the limitation of focusing on one compound class, and the motivation to test for differences across the IBPT.

*2) I am not quite convinced by the argument made regarding the IBPT depths and sediment age in cores 2D-13 and 4D-13. The jump from 51ka at 17m depth to 8.5ka at 15m in core 4D-12 is explained as either a period of low deposition or an erosional event (page 6). The IBPT in core 2D-13 is at ca. 16m (the "peak jump" in AD-12), the IBPT in core 4D-13 at about 9m. Neither the lithological nor the organic proxies allow for any easy core correlation. This is likely due to the different depositional settings and indicates that the depositional history is different, thus, also invoking a period of low deposition or and erosional event and expecting younger ages for cores 2D-13 and 4D-13 is not unequivocally justified. These two cores may not be affected by the same processes and sedimentation rates might be very different. Please provide additional age constraints (OSL or maybe 14C?) or discuss this much more carefully.*

Response: We agree that an alignment of the three cores is difficult and likely not possible based on the available information. Deriving age constraints on the time frames studied here is challenging due to the limitations of dating methods. Optically stimulated luminescence dating requires sampling and processing under dark room conditions. Only the OSL-dated samples described here have been processed accordingly; further OSL-dates can therefore not be retrieved for these cores. Radiocarbon dating, by contrast is only suitable for time frames of up to ca. 50 000 years and would be performed on macrofossils such as plant remains that can be affected by re-deposition, e.g. by thermokarst formation or fluvial transport, as suspected here. Nevertheless, we agree that additional [14]C dating could give valuable information specifically for the jump between 51 ka at 17 m depth and 8.5 ka at 15 m depth in core 4D-12; we will explore this option for upcoming studies. In response to the reviewer comment here, we adjusted the manuscript to not implicitly or explicitly suggest ages for the undated cores 4D-13 and 2D-13.

*3) you may want to consider additional lignin phenol data from the Buor Khaya Bay: Ulyantseva et al. (2018) The Molecular Composition of Lignin as an Indicator of Subaqueous Permafrost Thawing. doi: 10.1134/S1028334X1810029X.*

Response: Thanks for the hint, we were not aware of this publication. This study is now included in our comparison Figs. 4, 5b and 6 and the related Discussion, and the site of the core is marked in the Fig. 1 map. Nevertheless, we also point out that we do not agree with all aspects of the authors' interpretation, in particular with respect to attributing variability in S/V ratios to degradation after thaw.

***Specific comments***

*Page 2 l.3-5: it would read nicer to combine the subordinate clause with the remainder of the sentence.*

Response: We re-phrased the sentence.

*l.6: "destroyed by erosion" or eroded?*

Response: Changed to "*eroded*".

*l.7: "This process" - which process are you referring to? Erosion or inundation?*

Response: Both; we changed the text accordingly.

*l.8: add superscript to unit.*

Response: Changed as suggested.

*l.10: "due to the changing thermal gradient"?*

Response: One of the reasons for more rapid subsea than terrestrial permafrost thaw is the effect of the comparatively warm overlying ocean water; we re-phrased to clarify this point.

*Page 3 l.7: capitalize "delta".*

Response: Changed as suggested.

*l.12: please define the origin of Yedoma (vs. fluvial and alluvial deposits).*

Response: See above; the description of the study area and different preserved permafrost deposits has been extended to better outline differences between deposits and their relevance for this study.

*l.20: change header or include additional header after l.26. Most of the chapter does not reference the field work, but either IBPT deepening rates or laboratory methodology/sample processing.*

Response: The header has been changed to "*Field work and drill core description*".

*Page 4 l.11-12: move last sentence to introduction or discussion.*

Response: In the updated manuscript, lignin is introduced at the end of the Introduction, including also the motivation for choosing this biomarker (see above).

*Page 5 l.11: is OC not total OC? The distinction is made for N, but not OC.*

Response: We apologize for being unclear and changed to "*total organic carbon and total nitrogen*" throughout the text.

*l.30-31: please use common italics notation for phenols.*

Response: Changed as suggested throughout the manuscript.

*Page 6 l.1-9: please use common italics notation for phenols.*

Response: Changed as suggested throughout the manuscript.

*l.5: add information that all phenol concentrations were normalized to g OC.*

Response: Changed as suggested.

*Page 7 l.1-3: what is the threshold value to differentiate unimodal and bimodal distributions? Based on Fig. 2, 4D-12 also has substantial clay contribution (some 25% or so) in those intervals dominated by silt, but this does not qualify as a bimodal distribution?*

Response: Statistical properties of grain size distributions including modality were calculated applying the Gradistat v8 program (Blott and Pye, 2001). To derive the modes, the program first normalizes the fraction in each size class (as presented in Table S2 in this manuscript) by the difference between the logarithms (base 2) of the upper and lower size class thresholds. A mode is then defined as a local peak in the normalized fraction within a size class that reaches at least 15% of the overall largest peak of the respective sample. In the specific case of core 4D-12, the finer grained intervals typically contained both silt- and clay-sized grains but with one peak of the overall distribution in the silt fraction; an unimodal distribution was consequently identified. Nevertheless, although the parameters applied by the Gradistat program are frequently used, we agree that there is no universal definition of these statistical properties. In the updated manuscript, we added details of the calculations to the method description and adjusted the Discussion of differences in grain size distributions between cores to more critically consider the underlying data and calculations.

*l.9-27: it would be nice if the argument order in this paragraph was reversed, starting e.g. with the sentence in l.20 so the connection to the above paragraph is more obvious.*

Response: We followed the reviewer's recommendation and restructured the paragraph to better connect it to the paragraph above.

*l.19-20: does TC allow citing articles in review?*

Response: This paper has been published in the meantime and is now formally cited.

*l.22-23: more similar in comparison to?*

Response: Core 4D-13 is more similar to Ice Complex deposits with respect to grain size distribution than cores 2D-13 and 4D-12; we clarify this in the updated manuscript.

*Page 8 l.8: change to "normalized to"*

Response: Changed as suggested.

*l.15: what can we learn from comparison with active layer soils in Svalbard? That is a very different Arctic setting.*

Response: We agree and removed the comparison.

*l.18: O'Leary (1988) does not provide _13C values for plants in East Siberia.*

Response: The cited publication referred to the typical $^{13}C$ range of C3 plants; we nevertheless replaced this reference by studies from Siberia in the updated manuscript.

*l.18-21: please provide endmember values for the cited high latitude references.*

Response: Values have been added to the updated manuscript; see also Fig. 4.

*l.30-32: you are only looking at lignin phenols (not the molecular composition), so he data provide a very limited view on microbial degradability, since no other compound classes are assessed and lignin is a very refractory material to start with.*

Response: We agree and now address our motivation for focusing on lignin as well as the limitations of this approach in the updated manuscript.

*Page 9 l.18-22: while the acid to aldehyde ratio is used to determine the degree of aerobic decomposition, isn't the fact subsea permafrost in this area is anoxic below the SMTZ (which coincides with the IBPT; e.g. Winkel et al. 2018, Scientific Reports, doi:10.1038/s41598-018-19505-9) suggesting that this is rather a function of age/exposure time to aerobic conditions? Irrespective of transport distance and duration, if the OC from the boreal forests in the South was stored in soils prior to erosion and export, it is likely much more degraded.*

Response: We agree that decomposition after subsea permafrost thaw is unlikely to significantly affect acid-to-aldehyde ratios but that the variability in the cores rather reflects degradation before deposition (see also above). We also agree with the likely more degraded state of organic matter in southern than northern soils; we had not considered this before. Both points have been clarified in the updated manuscript.

*l.26-27: this statement contradicts with the previous statement arguing for a lower degree of lignin degradation in subsea permafrost samples vs. riverine and surface sediments (l.18-22). Also, based on Fig. 6, the heterogeneity in terrestrial deposits and surface sediments is large and within SD at least agrees with the concentrations in core 4D-13.*

Response: We understand that these statements might seem in contradiction. One possible explanation for the low lignin relative to organic carbon content in the studied subsea permafrost samples is lignin degradation under anoxic conditions; this would likely not result in increased acid-to-aldehyde ratios as pointed out also by Reviewer 1. Nevertheless, previous incubation studies do not suggest preferential lignin compared to bulk organic matter degradation under anoxic conditions, and other explanations have to be considered, for instance a high contribution of organic carbon from other sources than lignin-containing higher land plants. Both aspects are discussed in the manuscript. We further agree that variability both in terrestrial deposits and surface sediments as well as in core 4D-13 is high and that some samples of 4D-13 show lignin concentrations in a similar range as, for instance, Pleistocene Ice Complex deposits and fluvially influenced marine surface sediments. In response to the reviewer comments, we extended this part of the discussion to clarify the stability of acid-to-aldehyde ratios during anoxic lignin degradation and address the variability of lignin concentrations within subsea permafrost and other deposits. Note also the changes of Fig. 6 to box-whisker plots that simplify the comparison of subsea permafrost with other deposits.

*Page 10 l.2-3: I would argue that the data set is too limited to assess OC qualities. It allows to assess lignin quality; for OC quality, a comprehensive data set including various compound classes would be needed.*

Response: We agree. We did not mean to imply that one compound class is representative for the multitude of compounds that form organic matter; we address this limitation in the updated manuscript.

*l.8-14: this entire paragraph is phrased as if these measurements were performed on actual density fractions, which is not the case. This should be re-phrased to acknowledge that "samples with finer or coarser grain size distributions: : :"*

Response: We agree and re-phrased accordingly.

*Fig. 1 Please add inset boxes and panel IDs and increase the font size in all maps and the legend. Add contour lines and legends to the small maps on the right. These also miss coordinate systems.*

Response: Changed as suggested. See also further improvements to Fig. 1 based on suggestions of Reviewer 2.

*Fig. 2 What exactly is the reference point for the distances? The coastline?*

Response: The reference point is Cape North on Muostakh Island; this has been clarified in the updated manuscript.

*l. 4: change "has not been" to "was not"*

Response: Changed as suggested.

*Fig. 4 Please include abbreviations in figure caption or use unabbreviated labels. There are many more endmember values available for %OC and OC/TN, why are they not included? One example, several*

*marine sediments are referenced in chapter 3.2, which could be added to the plot. Why are the own data shown as box-whisker plots, but the reference data are not? Comparing medians and means is not straightforward.*

Response: First, abbreviations have been added to the legend. Second, we agree that we did not sufficiently explain the choice of deposits for comparison in the previous version of the manuscript; this has been clarified now (see also above). Following the reviewer's suggestion, we further extended the comparison with marine surface sediments in the manuscript, by including a wider range of previous publications to characterize marine sediments in the Buor-Khaya Bay that are strongly influenced by terrestrial carbon deposited by the Lena river, and by complementing these data with observations from the remainder of the Laptev Sea where fluvial influence is weaker (Figs. 4, 5, 6). Finally, in the previous version of the manuscript, means and standard deviations were plotted for data derived from previous publications since not all of those studies provided individual values. However, we agree that this restricted the comparability with our subsea permafrost data. For the updated manuscript, we therefore extracted the distribution of the original data from published box-whisker figures (where individual data were not published), permitting us to now use box-whisker plots for all parts of Figs. 4 and 6.

*Fig. 6 Please include abbreviations in figure caption or use unabbreviated labels.*

Response: Abbreviations have been added to the captions.

**Referee #2: Pavel Serov**

*The manuscript describes content and composition of organic matter within a recently thawed permafrost horizon encountered in 3 re-drilled offshore boreholes in the Laptev Sea. The results indicate that the bulk organic matter content is rather low, which resonates with previously published works. Lignin phenol concentrations are unexpectedly low, which the authors suggest is due to its preferential degradation, river transportation and re-deposition. Using grain-size distribution, the authors attempt to reconstruct depositional environment, which would support their interpretations of the sources of the organic matter. Interpretation of analysis points toward rather heterogeneous content, origin and composition of organic matter within a small study area (maximum distance between the boreholes seems to be _2 km) overriding any detectable effects of organic matter degradation. The data quality and analytical techniques used are beyond dispute. The text is clearly written, the very general title adequately reflects somewhat diluted focus of the paper, and the subject matter of the manuscript may fit the scopes of The Cryosphere journal. However, I found a few rather large-scale issues that prevent me from recommending this manuscript for publication at its current state.*

Response: We thank the reviewer for the critical feedback that has helped to considerably improve the manuscript. We agree that we have not been sufficiently clear on some aspects/context of the manuscript and address these issues in connection to the individual comments below.

*1. Lack of focus. It is not clear what this paper is about. When reading the Abstract and, partly, introduction I was sure it would tell a story of a recent thaw leading to organic matter degradation and greenhouse gas production. These first parts of text unambiguously point towards that. Yet, the results subchapters and discussion talk about depositional settings, permafrost evolution throughout the last*

*160 ka, origin and transportation of organic matter, etc. What all these findings have to do with the truly unique observations of rapid and recent thaw? In fact, investigating any permafrost core in the region (no matter if it is frozen or thawed, first time drilled or re-drilled) would reveal the same facts on sediment deposition, organic carbon contents, and nature of organic matter. In addition, I have to mention that the paper by Shakhova et al., 2017 covered substantial part of what is discussed in the manuscript.*

Response: We agree that the "stage setting" of the manuscript was not well aligned with the core investigational aspects of this study. While constraining the potential of subsea permafrost as a source of $CH_4$ to the atmosphere is a long-term aim and motivation for this line of research, $CH_4$ release is not the scope of the current study which takes some first steps at characterizing the organic matter in subsea permafrost. Based on the feedback by both reviewers and the editor, the manuscript has been substantially revised to better outline the actual motivation, scope and objectives, as detailed below. To avoid misunderstanding, we also changed the title to "*Organic matter at the thaw front of subsea permafrost on the East Siberian Arctic Shelf*".

**This study is motivated by the high demand for observational data on subsea permafrost carbon dynamics.** Subsea permafrost represents a potentially large and vulnerable organic carbon pool and might be a relevant source of greenhouse gases to the atmosphere. While organic carbon stocks and greenhouse gas emissions from terrestrial permafrost are increasingly well constrained, carbon stocks and dynamics of subsea permafrost remain highly uncertain due to a paucity of observational data from these hard-to-access systems. As a consequence, models of permafrost carbon dynamics currently have to ignore subsea permafrost (Schuur et al., 2015) or build on coarse assumptions (Lindgren et al., 2018).

To fill this knowledge gap, **this study provides a quantitative and qualitative assessment of organic matter at the subsea permafrost thaw front and compares it with terrestrial permafrost** deposited at similar times. Identifying terrestrial counterparts of subsea permafrost would permit future studies to make informed assumptions about subsea permafrost properties and responses, based on much better understanding of terrestrial permafrost.

These overarching study aims are addressed in three specific objectives, to
(1) **determine the age of subsea permafrost** and in particular of material at the current thaw front; this optically stimulated luminescence dating of the longest 56 m drill core provides the, to our knowledge, first age constraints of subsea permafrost and enables us to put subsea permafrost in the Buor-Khaya Bay into the stratigraphic context in the study region;
(2) **quantitatively and qualitatively describe organic matter at the subsea permafrost thaw front** based on organic carbon content and isotopic composition as well as biomarker properties; and
(3) **compare organic matter at the subsea permafrost thaw front with terrestrial permafrost** deposited over similar time frames, under a variety of environmental conditions in the study region.

We therefore do not agree that "*In fact, investigating any permafrost core in the region (no matter if it is frozen or thawed, first time drilled or re-drilled) would reveal the same facts on sediment deposition, organic carbon contents, and nature of organic matter*". While terrestrial permafrost is well described, observational data on subsea permafrost are extremely scarce. The differences in deposition age and thaw dynamics caution against assuming similar thaw responses of subsea and terrestrial permafrost. Terrestrial permafrost thaw occurs predominantly in the form of gradual active layer deepening affecting shallow Holocene deposits, and in the form of abrupt thermokarst formation and collapse of deeper and older deposits, often of Pleistocene Ice Complex Deposits that are particularly vulnerable due to their high ice content. Subsea permafrost thaw, by contrast, occurs gradually but at much greater depths, and affects

Pleistocene material of various ice contents. Identifying to what extent organic matter in currently thawing subsea permafrost can be compared to currently thawing terrestrial permafrost is a main long-term objective of this line of study.

Finally, we point out that it is not correct that the Shakhova et al. (2017) publication "*covered substantial part of what is discussed in the manuscript*". The Shakhova study contains the basic description of the cores also studied here, including grain size data, which we cite accordingly in the manuscript. However, this current study is unique in its focus on organic matter, and all data presented here except grain sizes are original to this study.

*2. Mismatch of the declared research question(s) and results-discussion. The manuscript stresses on importance of permafrost thaw for climate gas budgets. Therefore, degradation state of organic matter in the cores would be the most relevant part of the manuscript. However, on the second-to last page, the manuscript says that none of the analyzed parameters (org. carbon content, d13C, lignin phenol ratios, OC/TN ) is sensitive enough to capture the decomposition of organics matter. The other findings do not seem to have direct relation to the climate gas problem, or the manuscript does not explicitly show such connections. Perhaps, one way to approach answering the question of CO2 and CH4 release from the thawed strata would be to calculate potential maximum and minimum scenarios of greenhouse gas generation in these particular settings and assuming the estimated thaw rates. Knowing that 1.6 kg of OC m-2 thaw-out every year does not bring us to understanding the magnitude of associated climate gas production.*

Response: As espoused above, the original manuscript had a sub-optimal stage-setting/Introduction that was not well aligned with the actual scope of the study. The primary goal of this study was not to assess changes in degradation proxies with thaw but to improve the currently very limited understanding of subsea permafrost organic matter by comparison with a range of terrestrial permafrost deposits. This has been clarified in the updated manuscript (see also above), and the title has been changed to avoid misunderstandings.

*3. Watery conclusions. Throughout the manuscript, I noticed that several lines of evidence lead to somewhat empty conclusions, such as ´ntaken together these findings point at the accumulation of material of different origin at varying proportions..", or the ones saying that more research is urgently needed. I suggest cutting off all parts of the manuscript not leading to new/important results.*

Response: Together with sharpening the focus of the manuscript (see above), the Results and Discussion as well as the Conclusions sections have been adjusted to specifically address the implications of our findings for the re-casted study objectives.

*4. The above-mentioned issues brought me to a conclusion that I cannot say that the paper advances our understanding of how important the extremely fast permafrost thaw is for modulating CO2 and CH4 release. The manuscript does thoroughly describe several properties of the organic matter within investigated cores and provides a discussion of the sediment deposition and thaw history (which overlaps with the previously published paper by Shakhova et al., 2017). This does contribute to a growing number*

*of site-specific descriptive studies of OM composition within permafrost, but does not provide a transformative step towards understanding the consequences of rapid permafrost thaw.*

Response: Again, the introductory scope of the manuscript has been now revised to be much better aligned with the actual content. Hence, we understand and agree with the reviewer's assessment with respect to greenhouse gas releases. The goal of this work was not to quantify potential $CO_2$ or $CH_4$ release upon thaw; such an assessment would not be possible from organic matter properties alone. We do agree with the second point of this reviewer comment, that the study is contributing unique data to an extremely data lean situation for the cryosphere constituted by subsea permafrost. This is also the reformulated key objective of the study.

Hence, we are convinced of the unique value of this study. While terrestrial permafrost has been extensively studied and the parameters relevant for understanding its carbon dynamics are increasingly well constrained, even basic properties of subsea permafrost are highly uncertain due to a paucity of observational data from these hard-to-access cryogenic systems. Models of permafrost carbon dynamics consequently either ignore subsea permafrost (Schuur et al., 2015) or rely on coarse assumptions (Lindgren et al., 2018). We therefore also do not agree that "*This does contribute to a growing number of site-specific descriptive studies of OM composition within permafrost*"; see also above for differences in subsea and terrestrial permafrost thaw dynamics.

By providing observational data on organic matter at the subsea permafrost thaw front, our study represents an important step in our understanding of this potentially large and vulnerable pool, moving from rough assumptions to observation-based constraints.

*Page 1, line 29. "1.6 kg OC m$^{-2}$ year$^{-1}$". Place this number in the context. Is it a lot/not a lot compared to other regions? I believe this is important enough to be included in the Abstract.*

Response: We followed the reviewer's suggestion and now provide an estimate for organic carbon thaw-out in terrestrial permafrost by active layer deepening. Based on the average active layer depth and active layer deepening rate observed in recent decades in northeastern Siberia (Luo et al., 2016), the average organic carbon density at this depth in different permafrost soil types (Harden et al., 2012) as well as their relative distribution (Hugelius et al., 2014), we estimate an average thaw-out of 0.36 kg OC m$^{-2}$ year$^{-1}$ by active layer deepening in northeastern Siberia. Organic carbon thaw-out at our subsea permafrost site exceeds this estimate four-fold; this comparison is now included in the Results/Discussion section as well as in the Abstract.

*Page 2, line 9. "since the region is tectonically highly active". This reads as only tectonically active regions demonstrate a subseafloor heat flow. Rephrase.*

Response: We removed "*since the region is tectonically highly active*" to generalize the sentence.

*Page 2, line 17. "a strong supersaturation of CH4 in seawater above subsea permafrost". Supersaturation of the bottom water or surface water?*

Response: Supersaturation of $CH_4$ has been observed in both bottom and surface waters. Nevertheless, we removed this sentence when re-casting the Introduction.

*Page 2, line 32. "a very recent warm-up of subsea permafrost to the thaw point". What triggered this recent and pronounced thaw?*

Response: While large uncertainties regarding the mechanisms of subsea permafrost thaw remain (Nicolsky et al., 2012; Serov et al., 2015), it is likely that inundation by comparatively warm ocean water immediately started to gradually warm subsea permafrost (Shakhova et al., 2014), with thaw starting only later when permafrost temperatures had sufficiently increased. Permafrost thaw might have been further accelerated by salt water intrusion. We clarify this in the updated manuscript.

*Page 3, line 8. "The ESAS was exposed". Partly or entirely?*

Response: Reconstructions of the glacio-eustatic sea level suggest lower sea levels than today during the penultimate glaciation ca. 190-130 ka before present, followed by inundation of the ESAS during the Eemian interglacial ca. 130-110 ka before present, and then again lower sea levels and aerial exposure of large parts of the ESAS until the early Holocene sea level rise, with only the deeper, outer parts of the ESAS occasionally flooded in between (Romanovskii and Hubberten, 2001). However, this reconstruction does not consider tectonic processes. Paleontological records indicate that the current Oyogos Yar coast was located far inland during the Eemian and consequently point at considerable tectonic subsidence since the Eemian that shifted the relative coastline hundreds of kilometers towards the south (Kienast et al., 2011). We conclude that the inner ESAS including the Buor-Khaya Bay has likely been exposed to the atmosphere from at least ca. 190 ka until the Holocene inundation; we clarify this in the updated manuscript.

*Page 6, line 27. "reflects". No the best word in a sentence about optical method. "shows" would do better.*

Response: We agree and replaced the word as suggested.

*Page 7, line 6. "(Overduin et al., 2015a)". I suggest to add locations of mentioned cores on the map. It definitely has room for extra info.*

Response: Changed as suggested. Note also the additional improvements to Fig. 1 suggested by Reviewer 1 that have been implemented in the updated manuscript.

*Page 8, line 16. "conclude". Speculate.*

Response: Changed as suggested.

*Page 8, line 26. "at least partly". This needs explanations. Partly fluvial/alluvial and partly – what? Also, coarse grain size and low OM content may point towards a wide range of deposition environments.*

Response: We agree with the reviewer. Given the limited understanding of permafrost deposition at the core depth and during the time period targeted by this study, we refrain from speculating too much about potential deposition regimes and re-phrased more cautiously to: "*… the comparatively coarse grain sizes, low organic carbon contents and enriched $\delta^{13}C$ values of subsea permafrost at the current thaw front at the study site might indicate the contribution of material re-deposited by rivers.*"

*Page 10, lines 26-29. "Nevertheless, although organic matter degradation may alter several of the measured parameters (e.g., organic carbon content, OC/TN ratios, $\delta^{13}C$ values, lignin phenol ratios), none of these parameters is likely sensitive enough to capture the low rates of decomposition expected under the cold and anoxic conditions at the thaw front of subsea permafrost." Is it related to the study area, or to subsea PF thaw fronts in general? Aren't "cold and anoxic" conditions maybe expected everywhere along the thaw front worldwide?*

Response: We agree and removed "*subsea*" to generalize the statement.

*Page 11, line 10. "at least". Suggest rephrasing.*

Response: We agree and re-phrased to "*..., likely at least partly transported by rivers.*"

*Page 11, line 16. "Author contribution". Contribution of NS and AM are not mentioned.*

Response: Both mentioned authors contributed to writing the manuscript; this has been clarified in the updated manuscript.

**References**

Blott, S. J. and Pye, K.: Gradistat: A grain size distribution and statistics package for the analysis of unconsolidated sediments, Earth Surf. Process. Landforms, 26, 1237–1248, doi:10.1002/esp.261, 2001.

Hamilton, S. E. and Hedges, J. I.: The comparative geochemistries of lignins and carbohydrates in an anoxic fjord, Geochim. Cosmochim. Acta, 52(1), 129–142, doi:10.1016/0016-7037(88)90062-2, 1988.

Harden, J. W., Koven, C. D., Ping, C. L., Hugelius, G., David McGuire, A., Camill, P., Jorgenson, T., Kuhry, P., Michaelson, G. J., O'Donnell, J. A., Schuur, E. A. G., Tarnocai, C., Johnson, K. and Grosse, G.: Field information links permafrost carbon to physical vulnerabilities of thawing, Geophys. Res. Lett., 39(15), L15704, doi:10.1029/2012GL051958, 2012.

Hugelius, G., Strauss, J., Zubrzycki, S., Harden, J. W., Schuur, E. A. G., Ping, C. L., Schirrmeister, L., Grosse, G., Michaelson, G. J., Koven, C. D., O'Donnell, J. A., Elberling, B., Mishra, U., Camill, P., Yu, Z., Palmtag, J. and Kuhry, P.: Estimated stocks of circumpolar permafrost carbon with quantified uncertainty ranges and identified data gaps, Biogeosciences, 11(23), 6573–6593, doi:10.5194/bg-11-6573-2014, 2014.

Kienast, F., Wetterich, S., Kuzmina, S., Schirrmeister, L., Andreev, A. A., Tarasov, P., Nazarova, L., Kossler, A., Frolova, L. and Kunitsky, V. V.: Paleontological records indicate the occurrence of open woodlands in a dry inland climate at the present-day Arctic coast in western Beringia during the Last Interglacial, Quat. Sci. Rev., 30(17–18), 2134–2159, doi:10.1016/j.quascirev.2010.11.024, 2011.

Lindgren, A., Hugelius, G. and Kuhry, P.: Extensive loss of past permafrost carbon but a net accumulation into present-day soils, Nature, 560, 219–222, doi:10.1038/s41586-018-0371-0, 2018.

Louchouarn, P., Lucotte, M., Canuel, R., Gagné, J. P. and Richard, L. F.: Sources and early diagenesis of lignin and bulk organic matter in the sediments of the Lower St. Lawrence Estuary and the Saguenay Fjord, Mar. Chem., 58(1–2), 3–26, doi:10.1016/S0304-4203(97)00022-4, 1997.

Luo, D., Wu, Q., Jin, H., Marchenko, S. S., Lü, L. and Gao, S.: Recent changes in the active layer thickness across the northern hemisphere, Environ. Earth Sci., 75, 555, doi:10.1007/s12665-015-5229-2, 2016.

Nicolsky, D. J., Romanovsky, V. E., Romanovskii, N. N., Kholodov, A. L., Shakhova, N. E. and

Semiletov, I. P.: Modeling sub-sea permafrost in the East Siberian Arctic Shelf: The Laptev Sea region, J. Geophys. Res. Earth Surf., 117(3), 1–22, doi:10.1029/2012JF002358, 2012.

Romanovskii, N. N. and Hubberten, H. W.: Results of permafrost modelling of the lowlands and shelf of the Laptev Sea Region, Russia, Permafr. Periglac. Process., 12(2), 191–202, doi:10.1002/ppp.387, 2001.

Schuur, E. A. G., McGuire, A. D., Grosse, G., Harden, J. W., Hayes, D. J., Hugelius, G., Koven, C. D., Kuhry, P., Lawrence, D. M., Natali, S. M., Olefeldt, D., Romanovsky, V. E., Schaefer, K., Turetsky, M. R., Treat, C. C. and Vonk, J. E.: Climate change and the permafrost carbon feedback, Nature, 520(January 2016), 171–179, doi:10.1038/nature14338, 2015.

Serov, P., Portnov, A., Mienert, J., Semenov, P. and Ilatovskaya, P.: Methane release from pingo-like features across the South Kara Sea shelf, an area of thawing offshore permafrost, J. Geophys. Res. Earth Surf., 120, 1515–1529, doi:10.1002/ 2015JF003467, 2015.

Shakhova, N., Semiletov, I., Leifer, I., Sergienko, V., Salyuk, A., Kosmach, D., Chernykh, D., Stubbs, C., Nicolsky, D., Tumskoy, V. and Gustafsson, Ö.: Ebullition and storm-induced methane release from the East Siberian Arctic Shelf, Nat. Geosci., 7(January), 64–70, doi:10.1038/ngeo2007, 2014.

Shakhova, N., Semiletov, I., Gustafsson, O., Sergienko, V., Lobkovsky, L., Dudarev, O., Tumskoy, V., Grigoriev, M., Mazurov, A., Salyuk, A., Ananiev, R., Koshurnikov, A., Kosmach, D., Charkin, A., Dmitrevsky, N., Karnaukh, V., Gunar, A., Meluzov, A. and Chernykh, D.: Current rates and mechanisms of subsea permafrost degradation in the East Siberian Arctic Shelf, Nat. Commun., 8(May), 15872, doi:10.1038/ncomms15872, 2017.

Tesi, T., Semiletov, I., Hugelius, G., Dudarev, O., Kuhry, P. and Gustafsson, Ö.: Composition and fate of terrigenous organic matter along the Arctic land-ocean continuum in East Siberia: Insights from biomarkers and carbon isotopes, Geochim. Cosmochim. Acta, 133, 235–256, doi:10.1016/j.gca.2014.02.045, 2014.

Winterfeld, M., Goñi, M. A., Just, J., Hefter, J. and Mollenhauer, G.: Characterization of particulate organic matter in the Lena River delta and adjacent nearshore zone, NE Siberia - Part 2: Lignin-derived phenol compositions, Biogeosciences, 12(7), 2261–2283, doi:10.5194/bg-12-2261-2015, 2015.